# Natural Asphalts in Pavements: Review

Hugo Alexander Rondón-Quintana [1,*], Juan Carlos Ruge-Cárdenas [2] and Carlos Alfonso Zafra-Mejía [1]

1 Facultad del Medio Ambiente y Recursos Naturales, Universidad Distrital Francisco José de Caldas, Bogotá 110321, Colombia
2 Programa de Ingeniería Civil, Facultad de Ingeniería, Universidad Militar Nueva Granada, Bogotá 111071, Colombia
* Correspondence: harondonq@udistrital.edu.co

**Abstract:** Natural asphalts (NAs) can be an economical and environmental alternative in pavement construction. Most studies have investigated them as binder and asphalt mixture modifiers due to their high compatibility with conventional asphalts. In this article, some of the studies carried out on the use of NA in pavements are summarized and described in a chronological order. The main aspects described in the reviewed studies were the type of asphalt binder or modified mixture, the type and content of the modifier, the manufacturing processes of the asphalt or modified mixture, tests performed, and main results or conclusions. In general terms, NAs show better performance as binder and asphalt mixture modifiers in high-temperature climates. Additionally, they tend to improve water and ageing resistance. As main limitations, it is reported that NAs tend to negatively affect the workability and performance of asphalt mixtures in low-temperature climates. Finally, recommendations for future study topics are provided at the end of this paper.

**Keywords:** natural asphalt; gilsonite; asphaltite; Trinidad Lake Asphalt (TLA); rock asphalt; review

## 1. Introduction

Currently, oil continues to be one of the main natural resources used in the world as a primary energy source (representing more than 30% of total energy consumption). The largest proven oil reserves are in Venezuela (17.5%), Saudi Arabia (17.2%), Canada (9.8%), Iran (9.0%), Iraq (8.4%), and Russia (6.2%) [1], among others. As a result of the depletion of light and medium crudes, heavy oil and natural asphalts (NAs) have become an important source of raw materials to meet the growing demand for fuels and petrochemical products [2–5]. On the other hand, the existence of large quantities of NA mines worldwide and NA's relatively low price make it an alternative material of wide industrial use [6,7]. NAs could produce bitumen of various standards by compounding it with residues of heavy oil, reducing the duration of the process of bitumen production and energy consumption [8,9]. Furthermore, in the last decade, NAs have received great attention, as they can be used to modify binders and asphalt mixtures due to their high compatibility with conventional asphalts [10]. With these natural materials, the cost of modification could be lower compared with polymers, and the latter tends to separate from the asphalt binder, generating stability problems [11].

In Colombia, there is a great opportunity to use NAs as modifiers for asphalt mixes. The exact amount of NA in Colombia is unknown. However, there are open-pit mines where it can be legally exploited (e.g., Armero, Tolima; Norcasia, Caldas; Pesca, Boyacá; El Paujil, Caquetá). It is estimated that 26 mines exploit this material, most of them located in the Departments of Boyacá, Caldas, Caquetá, Cesar, Cundinamarca, Santander, and Tolima [12]. Natural asphalt mixtures tend to be sustainable, ecological, and economical materials, which can be extracted without the requirement of explosives. They do not need to be heated for use in road projects, reducing energy consumption and emissions. Additionally, in Colombia, the Ministry of Transportation issued Resolution 10099 on

27 December 2017, adopting the construction specifications [13] for using natural asphalt mixtures on roads with low traffic volumes (which make up about 70% of the country's road network) [14]. In Colombia, almost 20% of the road network is unpaved, and from what is paved, a little more than 50% is in fair to very poor condition [15]. All the above motivates the study and discussion of the use of these materials in Colombia.

The objective of this study was to conduct a chronological literature review on the study and use of NA in road pavements, especially to review their performance as modifiers of asphalt binders and asphalt mixtures, as well as the advantages and disadvantages of their use. The authors will take this review as a starting point for future studies on the use of NA in Colombia. In addition, this review will be a source of reference for students, academics, and researchers in civil engineering, pavements, roadways, geotechnics, materials, and related branches. Likewise, they can be consulted by entities that promote the development of environmentally sustainable techniques in road projects. It is important to note that, in the literature consulted, only one study reported a bibliographic review on the subject addressed in this article, which can be consulted in [16]. However, in that study, the central theme was the use of gilsonite in asphalt pavements. Unlike the study conducted by [16], the present article deals with all-natural asphalt materials, describes more research, and updates the information.

## 2. Natural Asphalts (NA)

NA could be subdivided into soluble and pyrobitumens (a natural hydrocarbon substance, solid, and different from bitumen because it is infusible and insoluble). Some natural bitumens are paraffin or petroleum wax (solid but soft, white or colorless wax, derived from petroleum, composed of saturated hydrocarbons, and used mainly to produce candles, polishes, cosmetics, and electrical insulators), pitch (has certain mineral contents), and asphaltites such as: (i) gilsonite or uintaite (solid black organic material, originating from the solidification of petroleum; carbon residue in the range of 10–20% by weight); (ii) grahamite or anthraxolite (similar to gilsonite, it is a bitumen-impregnated rock, which is the result of metamorphic changes in gilsonite; it differs by its high fixed carbon value—35 to 55%—and higher melting point); and iii) glance pitch or manjak (similar to gilsonite, but has higher specific gravity and carbon percentage—20 to 30%) [17,18]. Differences in the quality of NAs depend mainly on their depositional sources (differences in chemical and mineralogical compositions) [4]. If the NA reaches the ground surface, it forms bituminous springs, and if it remains underground, it will gradually solidify and oxidize, forming a solid and hard substance that is mineral asphalt (e.g., asphaltites) [7].

From the natural materials listed above, asphaltites are the most used material in pavements. These are usually heavy (4–18° API) [19,20], blackish, hard, exhibit high softening points [21], and are recognized as asphalt hardening materials due to their high asphaltene content [14,22,23]. The melting point of asphaltites is in the range of 200 to 315 °C [24]. Asphaltite is a rock of petroleum origin, formed from a liquid or semi-liquid asphaltic material present at depth, which is deposited in crevices, veins, cracks, and voids transported by pressure, gravity, and/or temperature during tectonic movements [25–28]. Afterward, the material solidifies and alters mainly due to the loss of light fractions and exposure to biodegradation processes, oxidation, water washing, and other chemical reactions that can cause an increase in molecular weight and a decrease in the atomic H/C ratio [24,29,30]. These materials are used in the production of paints and varnishes, road construction, automobile tire production, electrical insulation, and ink production. In the past, they were used as natural pitches for lining containers, floors, and walls, and, in general, as a moisture insulator. After some refining operations, synthetic gas [31], liquid fuel, ammonia, and sulfur can be obtained. In addition, they can be used in power plants for energy production [26,32,33]. They can also be used for heating in residential sectors because of their high calorific value [26,34], as fuel in industrial plants [35], and as an additive in sand molds used by the foundry industry [36]. As a result of the presence of carbon in its structure, it is a suitable adsorbent for a wide variety

of contaminants [32]. In the world, some valuable materials such as vanadium, nickel, and uranium can be produced from asphaltite ashes [37]. They are characterized by high heteroatom and metal content [3]. They contain some valuable crude metals, such as Mo, Ni, and V, and some radioactive metals, such as U and Th [38,39]. They are also a potential source of hydrocarbons [40] and could be precursors of high-tech carbon materials such as carbon fibers [41].

Within the asphaltites, gilsonite is perhaps the most studied in pavements. It was discovered in the early 1860s, and in 1888, Samuel H. Gilson and an associate established the first company to extract and market it on a commercial scale [42,43]. It is a natural fossil resource, similar to petroleum-derived asphalt with high asphaltene content [44–49]. It has a higher content of nitrogen than oxygen in its structure, giving it special properties of surface wetting and resistance to oxidation by free radicals [22]. It is a black, brittle material that can be crushed to powder [50]. It is a high-purity natural material with zero penetration at 25 °C, a softening point between 129 °C and 204 °C, and specific gravity at 25 °C of 1.04 to 1.06 [51]. Gilsonite is known for its ease of use, good affinity with asphalt, and low cost [16,43,51–54]. World gilsonite reserves are about 100 million tons and are found in countries such as the USA, Canada, Iran, Iraq, Russia, Venezuela, China, Australia, Mexico, and the Philippines [32,48,55]. The annual world consumption of gilsonite is more than 90 Mt, and the percentage of use in road paving is between 60 and 70% [47,48]. According to [56,57], the stiffness of gilsonite is about 50 times higher than that of conventional asphalt at room temperature. It has carbon content (above 80%) and a low H/C ratio, indicating a high degree of molecular condensation. It contains a large number of N, O, and S elements, which exist in polar functional groups [28] such as hydroxyl and a carboxyl group, which enhance the bonding of the aggregate and binder [57]. The physical and chemical properties of gilsonite range from a combination of refinery bitumen and carbon [32]. Another name given to gilsonites in some studies is rock asphalt—RA (extracted from mines or quarries depending on the type of deposit). The long coexistence time (millions of years) with nature gives them high stability and compatibility with asphalt binders. They are formed after billions of years of accumulation and changes under the combined action of heat, pressure, oxidation, catalysts, and bacteria [58–62]. They show economic and environmental benefits [63,64] and are mainly composed of coarse-grained sandstone completely impregnated with bitumen (10 to 35 wt% of the rock weight) [51]. Chemically, it is mainly composed of asphaltenes and other chemical compounds such as hydrogen, nitrogen, and oxygen. The asphaltene in asphalt rock, due to its polar functional groups, has anti-stripping, anti-oxidation, high adhesion, and temperature resistance properties [65]. Some asphaltene rock deposits can be found in Iran, Xinjiang, Qingchuan, Sichuan (China), and Buton Island (southeast Sulawesi, Indonesia). The latter are known locally as ASBUTON, and the deposits are estimated at 677 million tons with an average asphalt rock percentage between 13% and 20% [66].

Another widely used and studied NA pavement is Trinidad Lake (TLA), which is located on the island of Trinidad in the West Indies. It is perhaps the most well-known source of lake asphalt (47 ha, 87 m deep and contains approximately 10 million tons of asphalt; [67]). Finely divided minerals are dispersed throughout the bitumen [4]. The extracted TLA, when refined, presents soluble bitumen (53–55%), mineral matter (36–37%) and others (9–10%) [51,67]. The bitumen consists of maltenes (63%–66%) and asphaltenes (33%–37%). TLA minerals are composed of particles of various grades, as follows: Pass 200 (0.08 mm): 89.8%; Pass sieve 100 (0.17 mm): 8.0%; Pass sieve 80 (0.20 mm): 2.2% [68]. It is a material with a high softening point, high content of asphaltic matrix and resins, and a more gel-like structure [69]. According to [70], the mineral matter (named by them as ash) could improve the performance and high-temperature resistance of asphalt because of its small size, large surface area, and rough surface. It is widely used as waterproofing material in bridge decks or overlaps [71–73]. Due to the existence of mineral matter, high asphaltene content, and similar chemical composition with petroleum, TLA has good thermo-chemical stability, good oxidation and water resistance,

good adhesiveness, as well as good compatibility with asphalt [58,74–76]. In towns near Trinidad Lake in Trinidad and Tobago, they use it as a source of tourism [77].

Another type of natural asphaltic material is tar sand (also known as oil sand and bituminous sand). It is a sand deposit that is impregnated with a dense, viscous petroleum-like material called bitumen [78]. In situ, oil sand deposits are predominantly quartz sand surrounded by a fine thin film of water and fine aggregates with bitumen that fill the pore spaces between the sand grains. Inorganic materials in the oil sand composition constitute approximately 80% by weight, and bitumen and water constitute approximately 15 and 5%, respectively [79]. The composition and mineral content may vary with location and geologic condition [80]. Bitumen-impregnated sandstone deposits occur in a variety of stratigraphic and climatic environments [4].

## 3. Methods

The bibliographical review was carried out mainly in Scopus, Web of Science, ScienceDirect, Taylor and Francis, SpringerLink, American Society of Civil Engineering (ASCE), Google Scholar, and MDPI journals and scientific databases. Keywords such as "natural asphalt", "natural bitumen", "asphaltite", "Trinidad Lake Asphalt", "TLA", "gilsonite", "rock asphalt", "asphalt rock", "rock bitumen", "oil sand", "tar sand", and "BRA asphalt" were used for the search. Most of the studies conducted using natural asphalts are in the areas of petroleum and geology. Those related to pavements were read and analyzed (105 in total). The number of papers consulted per year of publication from 2005 to 2021 is shown in Figure 1.

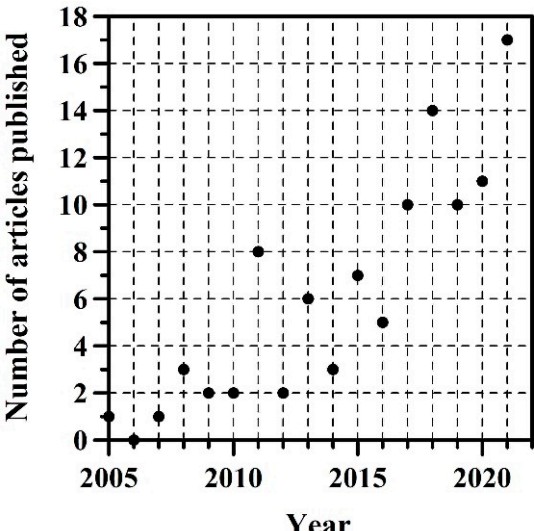

**Figure 1.** The number of articles consulted and published per year.

The review and analysis were carried out based on the type of NA used as modifier of asphalt binders and/or asphalt mixes. The main conclusions obtained from the studies were classified and described according to the physicochemical properties measured. The description of the review was arranged chronologically and is summarized and ordered as follows: authors, type of asphalt binder or modified mixture, type and content of modifier, manufacturing process of the asphalt or modified mixture, tests performed, main results, or conclusions. This information was summarized and organized in tables to make analysis easier.

The percentage of studies conducted on NA based on its use as a binder modifier, asphalt mixture, asphalt mastic, or other (uses in subgrade stabilization, sub-ballast, or concrete) is shown in Table 1. It also shows the most investigated type of NA. The major use has been as a modifier of asphalt binders. The most used NA has been RA and gilsonite-type asphaltites.

**Table 1.** Percentage of use of NA as modifier and type of NA studied.

| Modifier | Percentage (%) |
|---|---|
| Asphalt binder | 58.6 |
| Asphalt mixture | 37.9 |
| Asphalt mastic | 1.4 |
| Others | 2.1 |

| NA type | Percentage (%) |
|---|---|
| Gilsonite | 36.4 |
| TLA | 17.8 |
| RA | 37.4 |
| Oil sands | 0.9 |
| Asphaltite | 4.7 |
| Others * | 2.8 |

* This means that the type of NA is not clear.

During the description of the studies, the AC XX pen denotes Asphalt Cement (AC) with penetration of XX mm/10 (obtained at 25 °C, 100 g, 5 s; ASTM D5). "Conventional characterization" of asphalts refers to typical tests performed on asphalt binders such as penetration, softening point, ductility, viscosity, and specific gravity, among others. wt% refers to the percentage by weight.

## 4. Review

### 4.1. Gilsonite and Asphaltites

Liu and Li [52] modified PG 52-28 asphalt with gilsonite by a wet process in percentages of 3, 6, 9, and 12 wt%. The modification process is not clear. They performed conventional asphalt characterization, rheology using a DSR, Bending Beam Rheometer (BBR), and direct tension tests on long-term aged asphalts. The PG at high and low service temperatures increased from 52 °C to 70 °C and −28 °C to −22 °C, respectively. Gilsonite can improve the stiffness and permanent deformation resistance characteristics of mixtures used in hot climates. At low temperatures, the tendency of fatigue cracking and low-temperature cracking increases. Nevertheless, by adding a low gilsonite content (e.g., 3 wt%), the modified binder exhibits improved rutting resistance without compromising low-temperature cracking resistance.

Zhong et al. [81] modified PG 58 asphalt with two types of gilsonite (Xinjiang gilsonite and North America Gilsonite). The PG is not clear at low service temperature. The modification percentages were 5, 10, 15, and 20 wt%. The modification process is not clear. Viscosity and rheology with DSR and BBR tests were performed. Viscosity and PG at high temperatures increased with gilsonite content, improving stability and high-temperature resistance. Gilsonite reduced the low-temperature performance of the base asphalt. However, in small amounts, gilsonite would not have a considerable adverse influence on low-temperature performance. They recommended not exceeding the dosage of Xinjiang and North American gilsonite by more than 15 wt% and 8 wt%, respectively. The performance of asphalt modified with Xinjiang gilsonite is better than that modified with North American gilsonite.

Widyatmoko and Elliott [58] modified an AC 60–80 pen with 10 wt% gilsonite and 70 wt% TLA. They compared the response received by the modified asphalts with respect to the base AC and other asphalts modified with elastomeric and elastomeric polymers to which TLA was added at contents ranging from 15 to 70 wt%. The modification process is not clear. They performed conventional characterization of asphalts and rheology with DSR. In general terms, NAs increase the stiffness of the base asphalt.

Aflaki and Tabatabaee [82] modified eleven asphalt binders obtained from seven production plants in Iran (AC 40–50, 60–70, and 85–100 pen) with SBS, crumb rubber (CR), polyphosphoric acid (PPA), and gilsonite. The study performed rheological characterization, comparing the PG rating results of the modified asphalts with the 13 performance

grade binders required for specific climatic zones in Iran. Gilsonite was added at 2, 4, 7, 10, and 13 wt%. They did not perform conventional physical characterization tests on the modified asphalts. The particle size of gilsonite is not clear. Mixing was performed at 180 °C for 180 min at 4500 rpm. Gilsonite increases asphalt stiffness but deteriorates intermediate and low-service temperature cracking.

Suo et al. [83] performed indirect tensile stiffness modulus (ITSM) tests at 10, 20, 30, and 40 °C and indirect tensile fatigue (ITF) tests on three asphalt mixtures (gilsonite modified wearing course—GM-ACWC, stone mastic asphalt—SMA, and conventional asphalt concrete wearing course—ACWC) compacted in a gyratory compactor and subjected to short-term aging. The mixtures were designed using the Marshall method. A fatigue damage model was developed and a finite element analysis was performed to study the cracking resistance behavior of the mixtures. The base asphalt binder for the GM-ACWC mixture was AC 60/70 pen. The asphalt modification process is not clear. Conventional physical characterization results of the asphalt modified with gilsonite are shown. No rheological characterization of the modified asphalt was performed. They conclude that the longest fatigue life to crack initiation was obtained with the GM-ACWC mixture. However, the overall fatigue strength is lower when compared to the other mixes. They recommended the use of the GM-ACWC mixture where there is high traffic volume and high service temperatures.

Kök et al. [22] combined SBS (2, 3, 4, and 5 wt% of binder) and gilsonite (4, 5, 6, 7, 8, 9, 10, 11, 12, and 14 wt% of binder) as an additive to modify a B 160–220 asphalt binder (penetration, viscosity, or base asphalt properties are not clear). A SBS-gilsonite combination of 2–8, 2–10, 2–12, 2–14, 3–5, 3–7, 3–9, 3–11, 4–4, 4–6, 4–8, and 4–10 wt% was also used. Mixing was performed at a temperature of 180 °C and 1000 rpm for 1 h. They performed rheology tests with DSR and rotational viscosity (RV). They conclude that SBS and gilsonite increase asphalt stiffness, but the rate of this increase is higher when SBS is used. Additionally, the results show that 3% to 4% more Gilsonite is needed to replace 1% SBS. It is suggested that Gilsonite can be used as an alternative modifier to reduce the cost of producing and compacting the asphalt mixture in the field. At certain ratios of replacing SBS with gilsonite, a reduction in the viscosity of the base binder is observed, helping to increase the workability of the asphalt mixture during manufacture and reducing the compaction energy and the overall cost of pavement construction.

Ameri et al. [50] modified two asphalt binders PG 58-22 (AC 60 pen) and PG 64-22 (AC 89 pen) with gilsonite powder that passed the #50 sieve at 4, 8, and 12 wt% ratios. Initially, the base asphalt was heated to 140 °C and mixed with the gilsonite for 150 min at 150 revolutions per minute. Then, the mixing temperature was raised to 180 °C, and the process was repeated for another 30 min at 4500 rpm. The rheological properties of both modified asphalts were investigated. The gilsonite increased the stiffness of the asphalt, improving the degree of performance at high service temperatures. Low temperatures yield the opposite result.

Cholewińska and Iwański [84] modified an AC 50 pen asphalt with gilsonite in proportions of 5, 10, and 15 wt%. The process of addition and mixing of both materials is not clear. The scope of the tests was limited. They did not perform rheological characterization. A significant decrease in penetration and an increase in the viscosity and softening point of the modified asphalt as the gilsonite content increased were reported by the authors.

Ameri et al. [85] evaluated the performance of ethylene vinyl acetate (EVA) and gilsonite-modified asphalt binders according to the Superpave criteria. They only performed rheology tests using DSR, BBR, and RV. It is not clear what type of AC was used. The gilsonite powder passed through the #50 sieve and was added in quantities of 4, 8, and 12 wt%. Initially, the base asphalt was heated to 140 °C and mixed with the gilsonite for 150 min at 150 rpm. Then, the mixing temperature was raised to 180 °C. The process was repeated for another 30 min at 4500 rpm. Both EVA and gilsonite improve rutting resistance. The addition of EVA increases the fatigue cracking resistance, while the addition

of gilsonite decreases it. In general, the addition of gilsonite decreases the low-temperature cracking resistance.

Yilmaz et al. [86] used asphaltite as a filler (particles smaller than 0.075 mm) in an asphalt mixture. The proportion of asphaltite was 25, 50, 75, and 100 wt% of the filler. The type of asphaltite used is not clear. They used an AC 190 pen to make the mixture. Marshall, ITSM (15, 25, and 35 °C), Indirect Tensile Strength (ITS), and ITF (controlled stress and 25 °C) tests were performed on the mixture. The use of asphaltite as a filler improved the moisture damage resistance and fatigue life. The general trend of the mixtures was to increase Marshall stability, stability/flow ratio, tensile strength, and stiffness modulus with increasing asphaltite content.

Yılmaz et al. [87] used asphaltite as a filler (particles smaller than 0.075 mm) in hot-mix asphalts (HMA). They used PG 64-34 asphalt to make the mixtures. The type of asphaltite is not clear. In total, they produced six mixes: a control mixture with natural filler and five others where 1, 2, 3, 4, and 5 wt% of the natural filler were replaced by asphaltite. The mixture design was carried out by Superpave methodology. Marshall tests, resistance to moisture-induced damage test, and resistance to crack propagation (at 25 °C) tests were performed. The optimum proportion of asphaltite used as filler was determined to be 3 wt%. Asphaltite improved resistance to rutting, moisture damage, and crack propagation, especially at normal temperatures. However, they report that HMA mixes with asphaltite show more brittle behavior. They recommend the use of asphaltite as a filler to reduce the long-term effects of aging. They carried out an economic evaluation and concluded that the cost increase resulting from the addition of asphaltite is balanced by the decrease in total asphalt binder content in the design.

Hajikarimi et al. [88] used a BBR test to investigate the low-temperature rheological properties of a PG 58-22 asphalt binder when modified with CR (10, 14, 16 wt%) and gilsonite (2, 7, 13 wt%). They did not perform rheological characterization at high and intermediate service temperatures. The BBR test was performed at five different temperatures: −6, −12, −18, −24, and −30 °C. The mixing between the asphalt binder and gilsonite was carried out at 180 °C, 180 min, and 4500 rpm, while with CR it was performed at 170 °C, 240 min, and 5500 rpm, respectively. In conclusion, they determined that the modification of the binder with gilsonite reduces the anti-cracking capabilities of asphalt binders.

Yilmaz and Çeloğlu [89] used three different NA (TLA, Iranian gilsonite—IG, and American gilsonite—AG) and SBS as additives in the modification of a PG 58-34 asphalt binder. The modification was carried out at a temperature of 180 °C for 60 min at 1000 rpm. The authors mention that they used different proportions of the additives to modify the binder, but they do not mention them. Based on rheology tests using a DSR, they found the additive content necessary to obtain a minimum PG 70 (10% of AG, 9.5% of IG, 60% of TLA, and 3.8% of SBS). Accompanied by BBR tests, they found that the modified asphalts showed a PG of 70-34. On mixtures manufactured with the modified asphalts, they performed Marshall tests, resistance to moisture-induced damage test, ITSM test (20, 25, 30, and 35 °C), ITF test (25 °C), and cyclic creep test (50 °C). The best performance in most tests was achieved using TLA as a modifier. The highest resistance to moisture damage was obtained when IG and SBS were used. All additives increased the stiffness of the base binder. Mixtures using AG showed better performance compared to those using IG. NA was more effective in improving the performance of the mixes compared to SBS. However, when the efficiency per additive content (1%) of additive use was considered, SBS was found to be the most efficient additive.

Yilmaz et al. [90] used asphaltite as the filler in an HMA mixture using five proportions (1, 2, 3, 4, 5 wt%). They used PG 64-34 as the base asphalt. They performed the ITSM test (20 °C, 30 °C, and 40 °C), ITF test (25 °C), and cyclic creep test (50 °C) on the mixtures. The optimum content of asphaltite as filler was 3 wt%. They reported a significant increase in the stiffness under cyclic loading, permanent deformation, and fatigue resistance of HMAs when gilsonite is used as filler. The above reduces the asphalt content but increases the brittleness of the mixtures.

Sun et al. [91] modified an AC 60–80 pen with asphaltite from Xinjiang, China (4, 6, 8, 12, 16 wt%). Both materials were mixed at $160 \pm 5\ °C$ for 30 min at 500 rpm. The type of asphaltite is not clear. They performed conventional characterization and rheology tests with DSR (temperature sweep and time sweep) and BBR. Increasing the asphaltite content increased the complex shear modulus and decreased the phase angle, indicating an increase in rutting resistance. The fatigue life of the binders increased exponentially with asphaltite content. However, at low temperatures, asphaltite adversely affects its performance.

Rondón-Quintana et al. [23] evaluated the effect of dry and wet modification of an HMA mixture with a Colombian gilsonite. They performed rheology tests on AC 60–70 pen asphalt modified with 5, 10, and 15 wt%. Both materials were mixed at $160\ °C$ for 20 min. They performed rheological characterization using a DSR. They manufactured HMA mixtures using the Marshall method. They performed ITS, resilient modulus—RM (10, 20, 40 °C), and permanent deformation (60 °C) tests. Gilsonite increased the stiffness of the base asphalt and improved its performance at high service temperatures. In addition, increases in the strength and stiffness of the modified mixture were observed, despite the increase in void content. Higher mechanical strength and stiffness were obtained when the mixture was wet-modified with G/AC = 10%. Gilsonite helped to improve the rutting resistance of the HMAs. The HMAs with gilsonite showed similar resistance to moisture damage compared to the control mixture.

Guo et al. [92] modified a NA (99 pen at 25 °C pitch-type) with gilsonite (2, 3, 4, 5 wt%). They then fabricated three SMA blends. Neither the modification nor the fabrication process of the blends is clear. They performed conventional characterization on the NA and the modified one. On the blends, they performed a rutting test at high temperatures (dynamic stability) and the Marshall test. The result showed that the stiffness of the NA modified with gilsonite increased notably. Likewise, the Marshall and dynamic stability and the high-temperature performance of the mixtures improved with the addition of gilsonite.

Babagoli et al. [7] modified an AC 60–70 pen with an Iranian gilsonite passing the #200 sieve (5, 10, 15 wt%). Initially, both materials were mixed at 140 °C for 15 min at 150 rpm. Then, the speed was increased to 4500 rpm for 30 min. The main objective of this study was to determine the effect of gilsonite on the properties of SMA blends. They performed conventional characterization tests on the modified binders. They did not perform rheological characterization. They performed Marshall, ITS, moisture damage susceptibility, RM (25 °C), and rutting resistance (50 °C) tests on the mixtures. The use of gilsonite improved the performance of the SMA mixture in all the properties evaluated.

Djakfar et al. [93] evaluated the performance of a porous asphalt mixture using waste-recycled concrete and gilsonite. They used AC 60–70 pen as the base binder. The gilsonite contents used to modify the AC were 2, 4, 6, and 8 wt%. The modification process is not clear. The conventional porous asphalt mixture was mixed at 145 °C and the one containing gilsonite at 175 °C. Compaction was performed at 135 °C. A total of 90 Marshall specimens were prepared with varied asphalt content (5, 6, 7, 8, 9 wt%) and recycled aggregate (20, 40, 60, 80, and 100% replacement of virgin aggregate). They performed only permeability and Marshall tests. Gilsonite in the range of 8 to 10 wt% improved the Marshall properties of the porous mixture, particularly its stability, without significantly decreasing the permeability capacity of the mixture.

Huang et al. [94] modified a PG 64–22 asphalt binder with different additives (SBS, polyethylene—PE, PPA, gilsonite, bis-stearamide wax—EBS, and rubber processing oil). The main objective of the study was to evaluate the adhesion and self-healing properties of the modified binder. In the case of gilsonite, the additional content was 24 wt%. Binder bond strength (BBS) and DSR tests were performed on the modified asphalts. Hamburg wheel tracking device (HWTD) tests were performed on asphalt mixtures. The gilsonite modified the PG of the binder (changed from PG 64-22 to PG 82-16). This is an indicator of increased binder stiffness but decreased performance at low service temperatures. Additionally, it

improved the bond strength between base asphalt and aggregate, increasing the resistance to moisture damage.

Nasrekani et al. [95] modified a PG 58-22 asphalt binder with 5 and 10 wt% gilsonite (sieve #200). The binder was heated to 180 °C and mixed with the gilsonite for 1 h at 6500 rpm. Control and modified asphalt mixtures were manufactured using a Superpave gyratory compactor. They performed softening point and DSR tests on the asphalt binders. Dynamic creep tests (54.4 °C) were performed on the asphalt mixtures. The purpose of the study was to evaluate the performance of binders and asphalt mixtures at high service temperatures. Gilsonite increases the stiffness and elasticity of the asphalt binder. Both the asphalt binder and the modified asphalt mixture improved rutting resistance. They recommend the application of gilsonite-modified asphalt concrete for regions with high-temperature climates. A similar study, but evaluating resistance to moisture damage, was published a year later [53]. In the latter study, the chemical properties and functional groups of the modified asphalt were examined by FTIR. They also measured the indirect tensile strength of samples in a dry state and subjected them to freeze–thaw cycles. Samples modified with Gilsonite showed improved resistance to moisture damage. However, the improvement effect was not significantly different between the 5 and 10 wt% dosage of gilsonite.

Yilmaz and Yamaç [96] modified an asphalt binder, PG 52-28, with SBS, American Gilsonite (AG), and by combining both materials (5% SBS, 18% AG, 2% SBS + 13%AG, 3% SBS + 10% AG, 4% SBS + 6% AG). PG 52-28 was mixed with SBS and gilsonite for 60 min at a temperature of 180 °C at 1000 rpm. They evaluated the rheological properties of the modified asphalt binders. The combinations of SBS and AG were established with the criterion of obtaining PG 76-16 binders. Additionally, HMA mixtures were manufactured. Marshall, resistance to moisture-induced damage, ITSM (20, 25, and 30 °C), and ITF tests under stress-controlled (25 °C) were performed on the mixtures. The mixtures performed better in the Marshall and indirect tension tests when 18% AG + 3% SBS + 10% AG were used. For stiffness and fatigue life, the best results were obtained with 18% AG, 2% SBS + 13% AG, and 3% SBS + 10% AG. The highest resistance to moisture damage was obtained when using 5% SBS, 2% SBS + 13% AG, and 3% SBS + 10% AG. As a general conclusion of the study, they report that the use of SBS and AG combined provides greater benefits than using them individually.

Tang et al. [97] modified a PG 64-22 binder with gilsonite, PPA, and SBS, with the main objective of investigating rutting resistance. The contents of gilsonite to modify the binder were 4, 12, 16, 20, and 24 wt%. The gilsonite was mixed with the asphalt binder at 180 °C for 90 min. Rheological characterization (dynamic shear oscillatory and multiple stress creep recovery—MSCR tests) was performed on the modified asphalts. They also manufactured mixtures and performed a HWTD Test (60 °C). In general, the gilsonite-modified asphalts are much stiffer than those modified with PPA and SBS.

Themeli et al. [10] evaluated the molecular structure of asphaltite-modified asphalts during aging. For this purpose, they introduced a new parameter called ageing molecular distribution shift (AMDS). They used AC 50–70 pen and modified it with 5, 10, and 15 wt% asphaltite (upon modification, the asphalts changed to AC 35–50, 20–30, and 10–20 pen, respectively). Temperature and mixing time were 180 °C and 1 h, respectively. They performed rheological measurements and gel permeation chromatography (GPC). They compared the response obtained by the modified asphalts with asphalt binders of similar penetration derived from petroleum. They did not perform conventional asphalt binder characterization tests. According to the authors, asphaltite attenuates asphalt binder aging.

Jahanian et al. [43] modified an AC 60–70 pen with gilsonite (passed through sieve #200). The gilsonite contents were 2, 4, 6, 8, and 10 wt%. Initially, the AC was mixed with the gilsonite at 140 °C for 150 min at 150 rpm. Then, they heated it to 180 °C and mixed for 30 min at 4500 rpm. They did not perform conventional and rheological characterization of the modified asphalts. They manufactured asphalt mixtures with two different gradations and a gilsonite content of 4.6 wt%, but the process of choosing the gilsonite content and

manufacturing the mixture is not clear. They performed Marshall, ITS, RM, and DC tests. They also evaluated moisture sensitivity. The test temperatures are not clear. According to the authors, the addition of gilsonite to the AC reduces moisture sensitivity and significantly increases Marshall stability, RM, and rutting resistance.

Ameri et al. [98] modified two asphalt binders (PG 58-22 and PG 64-22) with gilsonite powder (passing sieve #50) (4, 8, 12 wt%). This was completed to investigate the rutting and fatigue resistance of the modified asphalt binders. Both materials were heated and stirred in a mixer for 140 min at 150 rpm until 140 °C was reached. Then, they were heated again for 30 min at 4500 rpm until the temperature reached 180 °C. Only rheology tests (MSCR and linear amplitude sweep—LAS) were performed. The addition of gilsonite improved the rutting resistance and fatigue properties (at low strain rates) of the asphalt binder. However, in the LAS test, as the shear deformation levels increased, the best fatigue performance was obtained by using the base asphalt binder.

Ren et al. [54] evaluated the effects of styrene–butadiene rubber (SBR) on conventional properties, rheological behaviors, storage stability, and compatibility of gilsonite-modified asphalt using DSR, BBR, FTIR, and fluorescence microscopy (FM). They used an AC 97 pen. Gilsonite-modified asphalts containing six dosages of gilsonite (5, 10, 15, 20, 30, and 40 wt%) were produced. The AC was first heated to 160 °C, and gilsonite was added. The sample was then stirred at 2500 rpm for 2 h. To prepare the gilsonite/SBR-modified asphalts, the AC was heated to 160 °C then mixed with 30% by weight of gilsonite and proportional SBR simultaneously before stirring at 2500 rpm for 2 h. The proportions of SBR were 2.5, 5.0, 7.5, 10, and 12.5 wt% to the mass of gilsonite-modified asphalt. Gilsonite and SBR increased viscosity, high-temperature performance, and rutting resistance. Nevertheless, the addition of gilsonite decrease low-temperature cracking and fatigue resistance, while the addition of SBR improves it. In addition, FTIR and FM tests show that both gilsonite and SBR interact with the asphalt physically rather than chemically during modification. They recommend 30 wt% gilsonite and 7.5 wt% SBR.

Sabouri et al. [99] modified PG 58-22 and PG 64-22 binders with 4, 8, and 12 wt% gilsonite. They also modified them with 3 and 5 wt% SBS for comparison. To make the gilsonite-modified asphalts, initially, the base asphaltic binders were heated to 140 °C and mixed with the gilsonite at 150 rpm for 15 min. Then, the speed was increased to 4500 rpm for 30 min to homogenize. Rheological characterization (DSR and LAS) was performed on the modified asphalts. Gilsonite increases the stiffness of the base asphalt and shows good performance in the LAS test at strain levels of less than 10%. They also manufactured asphalt mixtures and performed an FPB fatigue test (25 °C). In the case of the asphalt mixture with gilsonite, they used the asphalt modified with 12 wt% because it was the highest content evaluated and because it performed well in the LAS test. The gilsonite-modified asphalts showed performance improvement at medium and high temperatures. In the FPB test, the gilsonite-modified asphalt mixture achieved longer fatigue life.

Lv et al. [100] modified an AC 60–80 pen (PG 64-22) asphalt binder with five additives: SBS, branched SBS, gilsonite, high-density polyethylene (HDPE), and PPA. With gilsonite, the contents were 4, 8, 12, 16, 20, and 24 wt%. The AC and gilsonite were mixed at 180 °C for 90 min. They evaluated and compared three representative test methods that address asphalt moisture damage performance: surface free energy (SFE), BBS test, and HWTD (50 °C) test. Gilsonite increased the moisture damage resistance of the base binder. The optimum gilsonite content was 12 wt%.

Rondón-Quintana et al. [101] evaluated the performance of an HMA when the natural coarse aggregate (21 and 43 wt% of total aggregate) is replaced by a blast furnace slag (BFS) and an AC 60–70 pen is wet-modified with gilsonite (10 wt%). No characterization of the asphalt modified with gilsonite was performed since it was completed in previous studies. The temperature and mixing time of the AC with gilsonite were 160 °C and 40 min, respectively. Marshall, ITS, and Cantabro tests were performed. Moisture damage resistance was evaluated through the TSR ratio. The results showed that replacing the natural aggregate with BFS decreased the properties studied (resistance to monotonic

loading, moisture damage, and abrasion). However, these properties improved when gilsonite-modified asphalt was used. It is possible to obtain an HMA mixture with good mechanical properties when 21% (material retained on 1/2 ″ and 3/8″ sieves) by mass of the natural aggregate is replaced by BFS, AC is modified with gilsonite (10 wt%), and the mixing temperature is increased by 10 °C concerning the control HMA.

Rondón-Quintana et al. [102] evaluated the performance of a wet and dry modified porous asphalt mixture with a gilsonite. The base asphalt binder was AC 60–70 pen (PG 64-22). For the wet method, the temperature and mixing time of the AC with the gilsonite were 160 °C and 40 min, respectively. Characterization of the asphalt modified with gilsonite was not performed, since it was achieved in previous studies. Cantabro and Marshall tests were performed on the mixtures. They report a remarkable increase in stiffness under monotonic load and in abrasion resistance when using a G/AC = 10 wt% ratio by the wet method. In the dry method, an increase in the stability/flow (S/F) ratio of the Marshall test is observed, but the strength in the Cantabro test decreases. The results of the study suggest that Gilsonite as an asphalt modifier can be a material that can improve the stiffness and resistance to permanent deformation characteristics of porous asphalt mixtures.

Khanghahi and Tortum [103] modified an AC 89 pen with gilsonite powder by passing a #200 sieve (6, 12, 18 wt%). They mixed the AC and gilsonite at 180 °C for 60 min at 1000 rpm. They did not perform tests on the modified asphalt. They manufactured mixes with three different gradations and three different void contents using the modified asphalt. In addition, they added glass fiber (lengths of 3, 8, and 12 mm) in proportions of 0.1, 0.2, and 0.3 wt% of the aggregate. They performed a mixed Mode I/III fracture test and used the Taguchi method to determine optimum conditions for gilsonite and glass fibers as modifier materials in HMA. Based on the obtained results, they concluded that the experimental optimum conditions were fracture angle $\alpha$ 1/4 45°, experimental temperature $\Theta$ 1/4 $-15$ °C, air void content 4%, nominal maximum aggregate size 9.5 mm, gilsonite content 6%, and glass fiber weight and length 0.3% and 12 mm, respectively.

Rondón-Quintana et al. [104] substituted, in volume, part of the coarse fraction of a natural aggregate for BFS in an HMA. They used an AC 60–70 pen asphalt modified with gilsonite (10 wt%, obtained in previous studies). The AC was mixed with the gilsonite at 160 °C for 20 min. They performed conventional characterization and rheology tests with DSR on the modified asphalt. They manufactured HMA mixtures by increasing the mixing and compaction temperature by 10 °C concerning the control. They performed Marshall, ITS, RM (10, 20, 30 °C), permanent deformation (40 °C), and fatigue (20 °C) tests. They evaluated the resistance to moisture damage using the TSR ratio. Gilsonite significantly increased asphalt stiffness, increasing PG at high service temperatures (PG 58 at 70). Such an increase in stiffness caused HMA to significantly increase the Marshall quotient, RM, and resistance to permanent deformation. Gilsonite also increased the resistance to moisture damage and fatigue (but in the latter, the increases in strength were not statistically significant).

Mirzaiyan et al. [105] modified two binders (PG 58-22 and PG 64-22) with gilsonite (sieve #50) and SBS. The gilsonite contents were 4, 8, and 12 wt%. The SBS contents were 3 and 5 wt%. Initially, the two asphalts were mixed with the gilsonite at 140 °C and 150 rpm for 150 min. Then, the temperature was increased to 180 °C and mixed for another 30 min at 4500 rpm. On the modified asphalts, they performed conventional characterization, viscosity, and rheology tests using DSR, BBR, and FTIR. To evaluate the temperature susceptibility, they calculated three indexes: Penetration Index (PI), Activation Energy (AE), and Viscosity–Temperature Susceptibility (VTS). The addition of gilsonite and SBS increased the stiffness of the base and PG asphalts at high temperatures, without compromising the performance at low temperatures. Gilsonite increased the PI and AE indices. VTS increased with the addition of 12% gilsonite but decreased with the addition of 4% and 8%.

Sobhi et al. [49] modified an AC 62 pen with Sasobit (3 wt%) and gilsonite (5, 9, 13 wt%) to produce a Warm-Mixture Asphalt (WMA). Initially, gilsonite powder was mixed with the AC at 140 °C for 30 min at a speed of 2400 rpm. Then, the mixing temperature was

raised to 160 °C and sheared for another 30 min at 4000 rpm. Subsequently, the temperature was raised to 180 °C and the mixture was sheared for another 15 min at 4000 rpm. Finally, Sasobit was added and mixed for 20 min at a speed of 200 rpm at 140 °C. The manufacturing process of the WMA is not clear. Conventional characterization and rheology tests with DSR were performed on the modified asphalt. Microstructural evaluation using SEM was performed on the aggregate and gilsonite. DC test, ITS, RM, Marshall stability, and fracture tests (semi-circular bending samples) were performed on the asphalt mixtures. SFE measurement tests were also performed. The stiffness of the mixtures increased (improving rutting resistance) mainly due to gilsonite, and workability improved with the addition of Sasobit. However, Sasobit negatively affected the moisture susceptibility of WMA. The use of Gilsonite and Sasobit improved the thermo-mechanical properties and durability of the asphalt mixtures. The addition of Sasobit to the modified binder made it possible to reduce viscosity and produce asphalt mixtures at the same temperatures used for HMA production. At an intermediate temperature (25 °C), all the modified mixes showed higher fracture energy than the control mixture. In summary, Sasobit helps to decrease the production temperature, while gilsonite increases the resistance to moisture damage.

Zhou et al. [106] modified an AC (PG 64-22) with gilsonite (4, 8, 12, 20, 24 wt%), SBS (with sulfur stabilizer), CR, terminal blend (TB) CR, and HDPE. The mixing temperature of the AC and gilsonite was 180 °C. The modification process is not clear. BBS and FTIR tests were performed on the modified asphalt. On modified asphalt mixtures, they performed a Four-point beam (4PB) mixture healing fatigue test (25 °C). The manufacturing process of the mixtures is not clear. The design of the asphalt mixtures was carried out by the Superpave method, referencing a previous study. Fatigue tests were performed on mixtures manufactured with the recommended optimum dosages: 4.5% SBS-modified asphalt, 15% TB-modified asphalt, 24% gilsonite-modified asphalt, 8% HDPE-modified asphalt, and 18% CR-modified asphalt. Except for gilsonite, the other modifiers showed a negative effect on the initial bond strength of the asphalt. A proper dosage of gilsonite could improve the bond performance of the asphalt binder. Gilsonite at proper dosages could enhance the asphalt's self-healing property. The optimum gilsonite content was 12–20%.

Ameli et al. [57] modified an AC 95 pen with gilsonite (10, 20, 30, and 40 wt%) that passed through the #200 sieve and CR (adding 5, 10, 15, and 20 by weight of binder to 30% gilsonite). The percentage of 30 wt% gilsonite, over which CR was added, was obtained based on previous studies. Initially, the AC was heated to 170 °C to fluidize it. Then, CR was added and mixed for 30 min at 5000 rpm. Then, gilsonite was gradually added to the mixture and mixed for 30 min at 5000 rpm. On the modified binders they performed RV and rheology with DSR, MSCR, and LAS tests. They manufactured SMA blends, but their manufacturing process is not clear. They performed four-point beam fatigue (FPBF), RM (25 °C), ITS, DC (50 °C), and HWTD tests (60 °C) on the blends. Both gilsonite and CR stiffen the base binder as their contents increase, increasing rutting resistance. They also increase the binder and SMA resistance to fatigue cracking. The AC and the asphalt mixture modified with gilsonite and CR underwent better rheological performance at high and intermediate service temperatures. Both modifiers increased the ITS, RM, and Fracture Energy (FE) of the SMA. They recommended 30% gilsonite and 15% CR as optimum contents. A similar study to the previous one was published by Ameli et al. [107]. In the latter, the main objective was to evaluate the moisture damage resistance. For this purpose, in addition to the tests reported by Ameli et al. [57], they measured the TSR and performed the Texas boiling test. Based on the results obtained, the addition of CR increased the moisture damage resistance of SMA. The best moisture damage resistance was obtained when 30% Gilsonite and 20% CR were used.

Gopinath and Kumar [108] modified an AC 51 pen with gilsonite (10, 12.5, 15, 17.5 wt%). Initially, the AC was heated to 130–140 °C and gilsonite to 180 °C and then mixed for 90 min at 160 °C. Conventional and FTIR characterization tests were performed on the modified asphalt. Based on these tests, they recommended an optimum gilsonite content of 15 wt% and manufactured a High Modulus Asphalt Concrete (HMAC) designed

by the Marshall method. The mixing and compaction temperatures of the HMAC with gilsonite increased by around 26 °C and 35 °C, respectively. RM (35 °C) and immersion-type wheel rutting tests were performed. The gilsonite increased the OBC from 5.27% to 5.6%, increased the RM of the HMAC by 2.4 times, and improved the rutting resistance. They simulated the design of a pavement section and concluded that gilsonite could help reduce the required HMA thickness.

Zhou et al. [109] reported a study similar to the one published by Zhou et al. [106]. However, they measured additional aspects such as SFE on the modified binders and performed Cantabro and HWTD tests on the asphalt mixtures. The results show that gilsonite improved the bond strength and surface energy of asphalt. However, an excess of gilsonite (greater than 20%) would decrease this improvement. During modification with asphalt, gilsonite absorbs the light components (saturated non-polar and aromatic), forming a system with higher resin content. They mention that gilsonite can decrease the cracking resistance of asphalt at low temperatures. Gilsonite promotes the healing performance of asphalt. They recommended a gilsonite content between 12% and 20%.

Pahlevani et al. [64] studied the application of steel slag as a sub-ballast in railway construction. Railway sub-ballast layers require materials with high shear strength. Therefore, the researchers modified a slag with 1, 2, and 3 wt% gilsonite. On the slag with gilsonite, they performed the Los Angeles machine, direct shear, and California Bearing Ratio tests. The gilsonite significantly improved the shear strength properties of the slag. Gilsonite has no noticeable effect on the friction angle of the slag, although it improves its cohesion. The gilsonite-modified slag undergoes acceptable shear strength, making it a potential alternative as a sub-ballast material.

Zuluaga-Astudillo et al. [110] based on previous studies modified an AC 63.8 pen with a gilsonite (10 wt%) whose particles passed the #40 sieve. Both materials were mixed at 160 °C for 10 min at 3000 rpm. Conventional characterization tests were carried out on the modified asphalt. With the modified asphalt, they manufactured two HMA mixes with different gradations and substituted (by mass and volume) 21 and 24% of the natural aggregate with recycled concrete aggregate (RCA). They performed Marshall, ITS, RM (10, 20, 30 °C), permanent deformation (40 °C), fatigue (20 °C), and Cantabro tests on the mixtures. The gilsonite increased the stiffness of the base asphalt binder. If the replacement of natural aggregate by RCA was performed by volume, gilsonite and RCA could be used in HMAs for thick-asphalt layers in high-temperature climates and any level of traffic. By substituting by volume, the mixtures increased their strength under monotonic and cyclic loading (increased S, S/F, ITS, and RM, as well as resistance to permanent deformation and fatigue strength under controlled stress conditions), without increasing the asphalt binder content. They do not recommend the use of both materials in HMAs for thin-asphalt layers in low-temperature climates. They also do not recommend substituting natural aggregates for RCA by mass.

Nehrani et al. [111] used 5, 7, and 9 wt% of gilsonite as filler by the weight of the concrete cement. They also modified the concrete cement by 10, 15, and 20 wt% with rice husk ash (RHA). They performed flexural strength, tensile splitting strength, compressive strength, and abrasion tests on the concrete samples. The combination of the two additives reduced the compressive strength of the mixture. However, in some percentages, the tensile cracking and flexural strength of the concrete increased by 4 to 7%. The use of RHA and gilsonite has a positive effect on concrete abrasion.

Sobhi et al. [112] modified an AC 62 pen with gilsonite (5, 9, 13 wt%) and Sasobit (3 wt%). Initially, the AC was mixed with the gilsonite at 140 °C for 30 min at 2400 rpm. Then, the temperature was increased to 160 °C and mixing was continued for 30 min at 4000 rpm. Finally, it was mixed at 180 °C for 15 min at 4000 rpm. Sasobit was added at 140 °C and mixed for 20 min at 200 rpm. Conventional characterization tests, rheology with DSR, SFE test, storage stability test, and FTIR were performed on the modified asphalt. The addition of gilsonite and Sasobit increased the stiffness of the asphalt binder (improving performance at high temperatures and rutting) and produced a stable modified asphalt that

does not generate separation problems when stored. Sasobit helps improve the workability of gilsonite-modified asphalt. The gilsonite did not chemically react with the base asphalt and, combined with Sasobit, could help reduce moisture susceptibility in HMA and WMA. Asphalt with Sasobit probably decreases moisture damage resistance; however, gilsonite helps to increase it.

A summary of the review described in Section 4.1 is shown in Table 2.

**Table 2.** Gilsonite and asphaltite summary.

| Ref. | Asphalt Binder | NA Dosage (wt%) | Tests Carried out on Modified: | | Resistance to: | | | | | |
| | | | Asphalt Binder | Asphalt Mix | Rutting | Fatigue | Temperature Cracking | | Moisture | Aging |
| | | | | | | | Intermediate | Low | | |
|---|---|---|---|---|---|---|---|---|---|---|
| [7] | AC 60–70 pen | 5, 10, 15 | - | X | I | - | - | - | I | - |
| [10] | AC 50–70 pen | 5, 10, 15 | X | - | - | - | - | - | - | I |
| [22] | B 160–220 | 2,3,4,5% SBS + 4,5,6,7,8,9,10,11,12,14% G | X | - | I | - | - | - | - | - |
| [23] | AC 60–70 pen | 5, 10, 15 | X | X | I | - | - | - | S | - |
| [43] | AC 60–70 pen | 2, 4, 6, 8, 10 | - | X | I | - | - | - | I | - |
| [49] | AC 62 pen | 5, 9, 14 | X | X | I | - | - | - | I | I |
| [50] | PG 58-22 (AC 60 pen), PG 64-22 (AC 89 pen) | 4, 8, 12 | X | - | I | - | - | D | - | - |
| [52] | PG 52-28 | 3, 6, 9, 12 | X | - | I | - | - | S | - | - |
| [53] | PG 58-22 | 5, 10 | X | X | - | - | - | - | I | - |
| [54] | AC 97 pen | 5, 10, 15, 20, 30, 40 | X | - | I | D | - | D | - | - |
| [57] | AC 95 pen | 10, 20, 30, 40 | X | X | I | I | I | - | - | - |
| [58] | AC 60–80 pen | 10% G + 70% TLA | X | - | I | - | - | - | - | - |
| [81] | PG 58 | 5, 10, 15, 20 | X | - | I | - | - | D | - | - |
| [82] | AC 40–50, 60–70, 85–100 pen | 2, 4, 7, 10, 13 | X | - | I | - | D | D | - | - |
| [83] | AC 60/70 pen | Not clear | - | X | I | D | - | - | - | - |
| [84] | AC 50 pen | 5, 10, 15 | X | - | I | - | - | - | - | - |
| [85] | | 4, 8, 12 | X | - | I | D | - | D | - | - |
| [86] | AC 190 | 25, 50, 75, 100 of the filler | - | X | I | - | - | - | - | - |
| [87] | PG 64-34 | 1, 2, 3, 4, 5 of the filler | - | X | I | - | I | D | I | I |
| [88] | PG 58-22 | 2, 7, 13 | X | - | - | - | - | D | - | - |
| [89] | PG 58-34 | Not clear | X | X | I | - | - | - | I | - |
| [90] | PG 64-34 | 1, 2, 3, 4, 5 | - | X | I | I | - | D | - | - |
| [91] | AC 60–80 pen | 4, 6, 8, 12, 16 | X | - | I | I | - | D | - | - |
| [92] | NA (99 pen at 25 °C pitch-type) | 2, 3, 4, 5 | - | X | I | - | - | - | - | - |
| [93] | AC 60–70 pen | 2, 4, 6, 8 | - | X | I | - | - | - | - | - |
| [94] | PG 64-22 | 24 | X | X | I | - | - | D | I | - |
| [95] | PG 58-22 | 5, 10 | X | X | I | - | - | - | - | - |
| [96] | PG 52-28 | 18% G, 2% SBS + 13%G, 3% SBS + 10% G, 4% SBS + 6% G | X | X | I | I | - | - | I | - |
| [97] | PG 64-22 | 4, 12, 16, 20, 24 | X | X | I | - | - | - | - | - |
| [98] | PG 58-22, PG 64-22 | 4, 8, 12 | X | - | I | D | - | - | - | - |
| [99] | PG 58-22, PG 64-22 | 4, 8, 12 | X | X | I | I | I | - | - | - |
| [100] | AC 60–80 pen (PG 64-22) | 4, 8, 12, 16, 20, 24 | X | X | - | - | - | - | I | - |
| [101] | AC 60–70 pen | 10 | - | X | I | - | - | - | I | - |
| [102] | AC 60–70 pen (PG 64-22) | 10 | - | X | I | - | - | - | - | - |
| [104] | AC 60–70 pen | 10 | - | X | I | I | - | - | I | - |
| [105] | PG 58-22, PG 64-22 | 4, 8, 12 | X | - | I | - | - | S | - | - |
| [106] | PG 64-22 | 4, 8, 12, 20, 24 | X | X | - | I | - | - | I | - |
| [107] | AC 95 pen | 10, 20, 30, 40 | X | X | - | - | - | - | I | - |
| [108] | AC 51 pen | 10, 12.5, 15, 17.5 | X | X | I | - | - | - | - | - |
| [109] | PG 64-22 | 4, 8, 12, 20, 24 | X | X | - | - | - | D | I | - |
| [110] | AC 63.8 pen | 10 | - | X | I | I | - | - | - | - |
| [112] | AC 62 pen | 5, 9, 13 | X | - | I | - | - | - | I | - |

I: Increase; D: decrease; S: similar; G: gilsonite.

### 4.2. Trinidad Lake Asphalt (TLA)

Widyatmoko and Elliott [58] and Yilmaz and Çeloğlu [89] used TLA as an asphalt binder modifier. The description of both studies was presented in the previous chapter. The general conclusion in both studies is that TLA increases the stiffness of the base asphalt, and the best performance in most tests was achieved using TLA as a modifier.

Widyatmoko et al. [68] evaluated 18 mastic asphalts using a Dynamic Shear Rheometer (DSR). They also performed conventional physical characterization. They used as a control asphalt binder an AC 60–80 pen to which 70 wt% TLA was added. On other modified

asphalt binders (premixed with elastomers), TLA was also added in proportions ranging from 15 to 70 wt%. The process of adding TLA to asphalt binders to make mastic asphalts is not clear. The addition of TLA to the asphalt binders increased the stiffness of the base asphalt and mastic asphalts.

Cao et al. [72] modified a previously styrene–butadiene–styrene (SBS)-modified asphalt binder with TLA. The SBS content in the modified asphalt was 4%. TLA was added at contents of 15, 20, and 25 wt%. Both materials were mixed at 170 °C between 30 and 45 min. They performed conventional asphalt characterization tests. They made samples of cement concrete and asphalt mixtures. However, the specimen fabrication process is not clear. They performed shear, tensile, and water permeability tests. Composite-modified asphalt with TLA and SBS-modified asphalt can be used as a waterproofing bonding material for bridge deck pavement. The best performances were obtained when 20 wt% TLA was used.

Feng et al. [113,114] mixed an AC 64 pen with TLA at ratios of 40:60 wt%, respectively, at 170 °C for 30 min. The penetration of the modified asphalt declined to 27.8 mm/10. The manufacturing process of the mixture with the modified asphalt is not clear. They performed high-temperature stability, the capacity of moisture-resistance damage, and permeability tests. However, the description of these tests is not clear. Based on the results obtained, they conclude that the asphalt mixture modified with TLA has higher stability at high temperatures, higher resistance to moisture damage, and is more impermeable. Additionally, they mention that TLA ashes increase the fine content in the mixture.

Kim et al. [115] used an AC 60–80 pen modified with 30 wt% TLA as an asphalt binder. Then, part of the TLA in the control asphalt was replaced with SBS and antioxidants (0.4 to 5.0% by weight of the SBS modifiers) to improve fatigue cracking resistance. The optimum SBS-modified asphalt binder content was 8.3% by weight of the mixture, which includes 15% of TLA and 8.5% of SBS modifier by weight of the asphalt binder. They manufactured mastic asphalt mixtures at 240 ± 5 °C. Conventional characterization tests and visualizations using optical microscopy were performed on the modified asphalt. Four-point bending (FPB) fatigue tests (20 °C) and ITS tests (20 °C) were performed on the mixtures. According to the authors, SBS could reduce TLA contents in mastic asphalt mixtures by 50%, improving their physical and mechanical properties (especially crack resistance).

Liao et al. [116] modified an AC 60–70 pen with TLA (20, 25, 33, and 50 wt%). The TLA was heated in an oven at 200 °C for 2 h, and the AC was preheated to 160 °C in an oven for 1 h. TLA was then added and mixed with AC at 3000 rpm for 30 min. The objective of this work was to evaluate the effect of TLA on the rheological properties of modified asphalt. They performed conventional and rheological characterization (frequency sweep testing, multiple stress, and recovery testing). As for rheological properties, they measured complex modulus and phase angle, zero-shear viscosity (ZSV), and low and steady shear viscosity. The test results showed that TLA stiffens the asphalt by increasing viscosity and improving rutting resistance. However, high concentrations of TLA can make asphalt concrete very stiff and difficult to compact. It was recommended that the TLA concentration must be in the range of 20 to 33 wt%.

Yang and Xiaoning [69] modified two asphalts AC 70 and 90 pens with TLA at ratios of TLA/AC = 0:100, 10:90, 30:70, 50:50, 70:30, and 90:10. Both materials were mixed at 180 °C for 45 min. Conventional characterization and rheology tests were performed to measure the complex modulus and phase angle. The results showed that TLA increases the viscosity and stiffness of the base asphalts, improving high-temperature stability and rutting resistance. However, they mention that it is important to choose reasonable TLA content to meet engineering needs.

Li et al. [73] modified an AC 70–90 pen with TLA (5, 10, 20, 30 wt%). Both materials were mixed at 180 °C for 60 min. On the asphalts, they simulated aging processes with PAV (pressure aging vessel), TFOT (thin film oven test), and UV (ultraviolet). Conventional binder characterization tests were performed. Additionally, the chemical structure was analyzed by FTIR (Fourier Transform Infrared) and AFM (Atomic Force Microscopy) tests.

The addition of TLA increased the stiffness and viscosity of the system and may improve the aging resistance of the binder, forming a more chemically stable system.

Xu et al. [75] modified an AC 60–80 pen with TLA (10, 20, 30, and 40 wt%). Mixing of the AC and TLA was performed at 170 °C for 30 min at 4000 rpm. They performed conventional asphalt characterization and rheology tests (DSR and BBR). They simulated short and long-term aging. The results show that TLA improves performance at high temperatures (increases asphalt binder stiffness and rutting resistance) but degrades it at low temperatures (decreases cracking resistance). This is mainly due to the high asphaltene and ash content of TLA. In addition, TLA improves long-term aging resistance. They obtained an optimum addition content of TLA of 30 wt%.

Wang et al. [117] proposed the use of TLA and polyester fiber complex methods to obtain a high-modulus asphalt mixture. They used AC 70 pen as the base asphalt binder to produce SMA-13-type asphalt mixtures. The TLA contents were 15, 20, 25, 30, and 40 wt%. The polyester fiber content is 3 wt% concerning the aggregate. They added TLA and polyester fiber to the asphalt mixture by dry process (mixed polyester fiber, TLA, and aggregate for 90 s, then added it to the asphalt matrix mixing for 90 s, and finally mixed it with mineral powder for 90 s). The total mixing time was 4.5 min. The mixing temperature was between 165 and 175 °C, and the compaction temperature was 170 °C. They performed resistance to permanent deformation (60 °C), low-temperature crack resistance (−10 °C), water stability, and fatigue resistance (15 °C). They recommend 30% TLA + 3% polyester fibers. TLA significantly improved resistance to permanent deformation at high temperatures. The addition of 30% TLA increased the fatigue life of the asphalt mixture by 67%, but the fatigue life of the asphalt mixture decreased after this value.

Nciri and Cho [4] physicochemically characterized two NAs: Utah foam bitumen and TLA. They used elemental analysis, FTIR, thin-layer chromatography with flame ionization detection (TLC-FID), proton-nuclear magnetic resonance, spectroscopy (1H-NMR), and conventional asphalt characterization tests. They did not perform rheological characterization tests. They conclude that Utah oil sands have a high potential for oil production, while TLA is a suitable construction material for road pavement.

Xu et al. [74] modified an AC 78 pen with TLA and SBS. Initially, the AC was heated at 180 °C in an oil bath to obtain a flowable liquid. Then, a content of SBS (4 wt%) and sulfur powder (0.3 wt% to improve storage stability) was slowly added when the AC was stirred at 2500 rpm. This mixture was sheared at 4000 rpm for 1 h at approximately 180 °C. Then, the appropriate content of crushed TLA (30 wt%, as determined in previous studies) was gradually added. Finally, the mixture was stirred for 2 h at approximately 180 °C. On the modified asphalts, they performed TFOT, PAV, and UV aging simulation. They also performed softening point, viscosity, rheological characterization with DSR and BBR, and Thermogravimetric (TG) and FTIR characterization. According to this study, TLA and SBS show a combined effect in improving the anti-aging property of asphalt, which would extend the service life of asphalt pavement. They also improve rutting resistance. Adding SBS offsets the partial negative effect of TLA on the low-temperature performance of asphalt. TLA and SBS have little influence on the thermal stability of modified asphalt.

He et al. [118] modified two binders (AC 67 and 83 pens) with TLA (5, 10, 20, 30 wt%). The AC and TLA were mixed at about 175–180 °C for 60 min. They performed conventional characterization and rheology (DSR) tests. They also performed storage stability at high-temperature testing. TLA improved high-temperature stability and rutting resistance but decreased ductility. AC and TLA show good compatibility, and it is better when AC 67 pen was used. TLA generated greater increases in viscosity and stiffness of the AC 67 pen.

Liu et al. [119], based on previous studies, modified an AC 70 pen with TLA (5, 10, 20, 30 wt%) and SBR (2, 3, 4 wt%). Initially, SBR was slowly added to the AC and mixed at 160 °C for 20 min at 2000 rpm. Then, the speed was increased to 5000 rpm and TLA was added, mixing for another 30 min. They performed conventional and rheological characterization (DSR and BBR) on the modified asphalts. They simulated aging at TFOT

and PAV. The authors conclude that the modified asphalts could improve rutting resistance (by increasing initial stiffness) and prolong pavement life. However, TLA could degrade low-temperature flow properties, but this effect could be compensated for by SBR. They recommended combining 2% SBR and 20% TLA. Workability is negatively affected by the incorporation of TLA and SBS.

Fengler et al. [120] modified two binders (PG 64-22 S and PG 70-16 S) with TLA (15, 25, 50 wt%). The modification process is not clear. They simulated aging at TFOT, PAV, and UV. On the modified asphalts, they performed rheological characterization with DSR, MSCR, and LAS (at 19 °C) tests. They manufactured HMA mixtures, but the manufacturing process is not clear either. On the mixtures, they performed permanent deformation tests at 60 °C to calculate the FN and DM (4, 20, 40, 54 °C). The stiffness of the materials evaluated is directly proportional to the amount of TLA used, improving the resistance of the binder and the mixture to permanent deformation. Asphalts with TLA are more prone to cracking. In general, 25% TLA appears to perform the best.

Kołodziej et al. [76] modified an AC 42.8 pen with TLA (10, 20 wt%) to study the effect of short-term aging on its physical and rheological properties. Initially, the AC and TLA were heated at 160 °C for 1 h to fluidize them. They were then mixed at 3000 rpm for 5 min. They performed conventional characterization and rheology (DSR) tests and measured the ZSV. Simulated short-term aging of the TLA-modified binders did not worsen, nor did it reduce, the resistance to permanent deformation. They recommended 20% as the optimum TLA content; however, they mention that at that percentage, the modified asphalt is more susceptible to short-term aging. TLA improved rutting resistance.

Zhou et al. [121] manufactured a hot-mixture recycled asphalt mixture (RHMA) using an AC 64.2 pen and reclaimed asphalt pavement (RAP) in proportions of 50 and 100 wt% of the aggregate. Initially, they tested 11 types of asphalt binder samples: 100% virgin asphalt (VA), 100% TLA, 40% TLA + 60% VA (they called this mixture HA), 100% aged VA (heated to 5, 12, 19, 26 h), and 50% asphalt heated to 5, 12, 19, 26 h + 50% HA. Initially, VA and TLA were heated, respectively, to 150 °C and then blended for 15 min at 165 °C. Conventional characterization, rheological (DSR), FTIR, and AFM observations were performed on these asphalts. The manufacturing process of the mixture is not clear. DM (15 °C), high-temperature rutting test (60 °C), and 3-PB fatigue test (15 °C) were performed on the mixture. The addition of TLA leads to a change in the ratio of asphalt binder components, but no new functional groups are produced. The addition of TLA decreases the number of bee-like structures, forming a more stable system. The low-temperature performance of the recycled asphalt binder decreases, while at high temperatures it increases and fatigue damage develops rapidly. The RHMA mixture with 50% RAP has similar properties to the control mixture without RAP. When 100% RAP was used, the high and low-temperature performance and fatigue resistance of RHMA are better than the control mixture.

Sun et al. [122] manufactured a Gussasphalt (GA) type mixture with an AC 62 pen and AC 92 pen, China Qingchuan RA, TLA, three kinds of basalt aggregate, and a limestone mineral filler. On modified asphalts (20 wt% TLA for AC 62 pen and 12.5 wt% QRA for AC 92 pen), conventional characterization tests were performed. The GA manufacturing process is not clear. Differential Scanning Calorimetry (DSC) tests, SEM observations, and X-ray Computerized Tomography (CT) were performed. RA improved stability and performance at high and intermediate service temperatures. The low-temperature crack resistance of the mixture with TLA is better than that with RA.

A summary of the review described in Section 4.2 is shown in Table 3.

**Table 3.** TLA summary.

| Ref. | Asphalt Binder | TLA Dosage (wt%) | Tests Carried out on Modified: | | Resistance to: | | | | | |
|---|---|---|---|---|---|---|---|---|---|---|
| | | | Asphalt Binder | Asphalt Mix | Rutting | Fatigue | Temperature Cracking | | Moisture | Aging |
| | | | | | | | Intermediate | Low | | |
| [58] | AC 60–80 pen | 10% G + 70% TLA | X | - | I | - | - | - | - | - |
| [68] | AC 60–80 pen | 15 to 70 | X | X | I | - | - | - | - | - |
| [69] | AC 70, AC 90 | TLA/AC = 10:90, 30:70, 50:50, 70:30, 90:10 | X | - | I | - | - | - | - | - |
| [89] | PG 58-34 | Not clear | X | X | I | - | - | - | I | - |
| [72] | SBS-modified | 15, 20, and 25 | - | X | - | - | - | - | I | - |
| [73] | AC 70–90 pen | 5, 10, 20, 30 | X | - | I | - | - | - | - | I |
| [74] | AC 78 pen | 30 | X | - | I | - | - | - | - | I |
| [75] | AC 60–80 pen | 10, 20, 30, 40 | X | - | - | - | - | D | - | I |
| [76] | AC 42.8 pen | 10, 20 | X | - | I | - | - | - | - | S |
| [113] | AC 64 pen | AC/TLA = 40/60 | - | X | I | - | - | - | I | - |
| [114] | AC 64 pen | AC/TLA = 40/60 | - | X | I | - | - | - | I | - |
| [115] | AC 60–80 pen | 30 | - | X | I | - | - | - | - | - |
| [116] | AC 60–70 pen | 20, 25, 33, 50 | X | - | I | - | - | - | - | - |
| [117] | AC 70 pen | 15, 20, 25, 30, 40 | - | X | I | I | - | - | - | - |
| [118] | AC 67, 83 pens | 5, 10, 20, 30 | X | - | I | - | - | - | - | - |
| [119] | AC 70 pen | 5, 10, 20, 30 | X | - | I | - | - | D | - | I |
| [120] | PG 64-22 S, PG 70-16 S | 15, 25, 50 | X | X | I | - | - | D | - | - |
| [122] | AC 62 pen, AC 92 pen | 20 | X | - | I | - | - | - | - | - |

I: Increase; D: decrease; S: similar.

### 4.3. Rock Asphalt (RA)

Lu et al. [123] modified an AC 60 pen with 2, 4, 6, and 8 wt% natural rock asphalt (RA). The addition and mixing process of both materials is not clear. The composition of the RA and its effect on the properties evaluated are also unclear. Conventional asphalt characterization tests were performed on these mixtures. They also performed rheology tests using DSR and BBR and a rotational viscometer. The addition of RA increased the stiffness of the base asphalt, improving the stability at high temperatures. Nonetheless, they report an adverse influence at low service temperatures.

Xiao-jin et al. [65] evaluated the laboratory performance of an asphalt mixture that was modified with different dosages of a RA. They used an AC 70 pen and modified it with 5, 7.5, 10, and 12.5 wt% of RA. The process of modifying the AC is not clear. The manufacturing process of the modified asphalt mixture and the optimum percentage of RA used are also unclear. Conventional characterization tests were performed on the modified asphalt. The RA increased the stiffness of the base asphalt. Multiple creeps (flow number at 50 °C), freezing–thawing split, split test at low temperature ($-10$ and $-15$ °C), and dynamic modulus (DM; 50 °C) tests were performed. An SBS-modified mixture (4 wt% was the SBS content) was used as a control mixture for comparison. RA can improve the performance of asphalt mixtures at high temperatures and the resistance to moisture damage. The optimum recommended RA content was 5.0 to 7.5 wt%.

Hadiwardoyo et al. [66] performed skid resistance tests using a skid resistance tester and a British Pendulum Tester. An AC 67.75 pen was used. This asphalt binder was mixed with a RA called Buton Natural Asphalt (BNA) at percentages of 20, 25, 30, 35, and 40 wt%. The process of mixing the BNA with the binder is not clear. The manufacturing process of the asphalt mixtures is also unclear. The tests were conducted under wet surface conditions at temperatures between 30 and 55 °C. The BNA increased the skid resistance and the asphalt penetration index.

Li et al. [59] prepared Fine Aggregate Matrix (FAM) mixtures, incorporating three types of RA from Buton Island, China-QC, and Iran-UM. The dosing process was carried out by replacing the aggregates and the binder in weight and volume of the RA in the final mixture. They used PG 64-22 asphalt. Based on previous studies, 8% RA, QC, or UM and 20% Buton RA by weight of the total binder content were added to the mixture. Frequency–temperature sweep tests using DSR and creep–relaxation tests using BBR were performed. Tensile strengths of the FAMs at low temperatures were measured.

The commercially obtained RA powder was first mixed with oven-dried limestone aggregates and then with PG 64-22 binder. The mixture was then subjected to a short-term aging (STOA) process to be compacted in the Superpave gyratory compactor. Additionally, another set of loose mixes was placed in an environmental chamber at 60 °C for 30 days before compaction to simulate long-term aging (LTOA). The addition of RA increases the stiffness, slightly reduces relaxation potential at low temperatures, and increases the aging resistance of the FAM. QC exhibits the highest dynamic shear modulus value, followed by UM and RA Buton. The rheological properties of the FAM blends with RA were less affected by long-term aging compared to the control blend. RA had a slightly negative influence on the low-temperature performance.

Karami and Nikraz [124] manufactured an asphalt mixture using an AC (170 Pa.s viscosity at 60 °C) and BRA (20 wt%). They designed the mixes using the Marshall method and kept the optimum binder content (OBC) constant in both the control mixture and the mixture manufactured with BRA. The BRA was added to the mixture along with the AC. They describe the mixed manufacturing process. They performed repeated flexural bending tests at 20 °C. The BRA helped to improve the fatigue strength of the mixture. Mixtures with BRA showed more elastic and less viscous behavior than unmodified mixtures.

Suaryana [125] evaluated the performance of an SMA when the filler (9.5 wt% of the aggregate) was replaced with an RA (ASBUTON) and cellulose. They used an AC 60–70 pen to manufacture the SMA. They performed DM (4, 20, 45 °C), permanent deformation (60 °C), and fatigue strength (20 °C) tests. The results obtained showed that ASBUTON can prevent asphalt drain down and increase the filler ratio. The addition of ASBUTON increases the dynamic stability of SMA. In terms of fatigue resistance, both additives generate SMA mixes with the same performance. The DM master curve indicates that SMA with ASBUTON is relatively stiffer at high temperatures, but relatively less brittle at low temperatures.

Shi et al. [126] used a material they named Budon natural RA—BRA (after its origin), which consists of 25% to 30% bitumen and 70% to 75% natural mineral (with a maximum particle diameter of 1.18 mm). They performed an X-ray diffraction (XRD) test, X-ray fluorescence (XRF) spectrometry, and micromorphology analysis on the BRA. It appears that they added 20% BRA in an asphalt matrix with an AC 70 pen and performed conventional asphalt characterization tests. The mixing process of BRA with AC is not clear. They performed tests (Marshall, dynamic stability, thawing freeze tear strength ratio, low-temperature performance experiment) on mixtures using BRA, but the mixture manufacturing process is not clear. The methodological design of the experimental phase is not clear either. According to the authors, BRA has (i) high asphaltene content and strong cross-linking, (ii) plentiful micropores and high alkali content, (iii) good aging resistance, and (iv) helps to improve the mechanical performance of the mixture at high and low temperatures, as well as water resistance.

Zha et al. [127] used extraction and combustion methods to study the limestone mineral (LM) and NA of a BRA whose NA approximate content was 25%. They made SEM observations of the BRA. On the LM, they performed gradation, specific surface area test, and XRD analysis. LM can be used to replace mineral powder in asphalt mixtures due to its large specific surface area and porous structure. On the NA, they measured the saturated, aromatic, colloid, and asphaltene fractions. They also implemented element composition analysis and molecular weight determination. NA is a hard and natural asphalt whose asphaltene contents and aromatic ring structure are high, and its asphaltenes and colloids have a high molecular weight. NA has a stable colloidal structure and good compatibility with petroleum asphalt. NA is strong in micellar polarity, molecular association ability, and resistance to oxidative radicals and infiltration. Its viscosity, hardness, and softening point are high, but ductility is poor. To evaluate the effect of BRA as a modifier of road petroleum asphalt, 20% NA was mixed into the AC 70 pen. The modification process is not clear. They performed conventional and rheological characterization and analysis of infrared absorption spectra and thermal stability of BRA-modified asphalt. BRA can improve the strength of asphalt binders at high

service temperatures. However, it is poor in rheological properties at low temperatures. It has high nitrogen content and good adhesion to aggregate, so it can improve the resistance to moisture damage. On the other hand, BRA is a material devoid of undesirable materials, such as waxes, present in petroleum-derived asphalts.

Li et al. [128] evaluated six types of mixing processes of BRA with aggregates, AC 70 pen, and mineral powder to produce an asphalt mixture. They used 3 wt% BRA. In all processes, the total mixing time was 240 s and the aggregate's heating temperature was 185 °C. BRA is room-temperature and the base asphalt was heated to 155 °C. Marshall, rutting resistance, ITS, and water stability of mixtures tests were performed. They also performed contact angle measurement, inverse gas chromatography–IGC test, and analysis of the work of adhesion. Based on the results obtained, it seems that the traditional way of mixing BRA with the other components to make asphalt mixtures (first, to mixture aggregates with BRA, and then to mixture them with heated base asphalt and mineral powder) is not the most suitable for this purpose.

Zhong et al. [60] used a RA from Xinjiang (China) to modify an AC 87 pen, employing contents of 5, 10, 15, and 20 wt%. Initially, the AC was preheated at 150 °C and the RA was melted at 160 °C. Then, both materials were mixed at a temperature of 175 °C for 1 h, and subsequently, they were mixed for 15 min. Rheological characterization (DSR and BBR tests) was performed on the modified asphalts. They also manufactured asphalt mixtures designed by the Marshall method. The mixes were subjected to ITS tests to determine the tensile strength ratio (TSR), HWTD test (60 °C), three-point bending beam (3-PB) fracture test (−10 °C), and 3-PB fatigue test (20 °C). The addition of RA significantly improved the high-temperature performance of binders and asphalt mixtures, especially when a higher dosage was used. It also improved ITS, moisture damage resistance, and fatigue resistance. However, the addition of RA showed slight adverse effects on low-temperature performance.

Zhong et al. [129] modified two asphalt binders (AC 67 and 87 pen) with an RA from Xinjiang (China). The RA contents were 5, 10, 15, and 20 wt%. Initially, they preheated the AC and RA to 150 °C. Then, they mixed them at 175 °C for 30 min and kept them in an oven for 1 h, and finally liquefied them at 175 °C for 15 min. They performed conventional characterization and rheology tests (DSR, BBR) on the modified asphalts. They made mixtures and performed Marshall, water stability, and low-temperature bending (−10 °C) tests. The results show that RA is useful for improving asphalt performance at high temperatures (especially when the RA content increases), but it can affect performance at low temperatures. It also improved water resistance and fatigue resistance. The optimum RA content range is between 5 and 15 wt%.

Liu et al. [130] used an AC 70 pen and mixed it with BRA (contained approximately 25 wt% asphalt and 75 wt% mineral powder). Based on the recommendation of the BRA supplier, the dosage of BRA mineral powder was determined to be 2 wt% over the total mass of mineral aggregates. That is, BRA was mixed with the remaining aggregates and limestone mineral filler in a ratio of 2.67:98. The optimum asphalt content according to the Superpave design was 3.9% (including the asphalt in BRA). Mixing and compaction temperatures of BRA-modified Superpave mixture were 170–175 °C and 150–160 °C, respectively. They performed the HWTD test (60 °C), bending beam test (−10 °C), ITS ratio test, DM test (20, 30, and 40 °C), and flow number test (30, 40, 50, and 60 °C). The high-temperature performance and moisture damage resistance of the BRA-modified Superpave mixture improved, but the low-temperature performance decreased slightly compared to the control mixture.

Shi et al. [131] analyzed the rheological properties of 16 asphalts with different nano-silica and Qingchuan RA (QC) contents by univariate analysis and variance analysis. They modified an AC 60–70 pen asphalt with QC contents of 2, 4, 6 wt% and nano-silica of 1, 2, 3 wt%. They initially mixed the AC, QC, and nano-silica at 170 °C for 30 min at 5000 rpm after artificial stirring for 20 min. They performed RV tests, rheology with DSR and BBR, and Scanning Electron Microscope (SEM) observations. The optimum QC and nano-silica contents were 6% and 1%, respectively. Both materials increased the

viscosity and stiffness of the base binder, improving the resistance to deformation at high temperatures. The material that provided the highest stiffness was QC. On the other hand, the modified asphalts show slightly lower performance at low service temperatures as the additive content increases. They mention that it is not cost-effective to use nano-silica alone to improve the anti-rutting performance of asphalt. Modified asphalt is more suitable for high-temperature areas or transport of heavy loads, whereas it is not suitable for cold weather under $-16\ °C$.

Cai et al. [132] modified an AC 60–70 pen with nano-silica (NS), RA, and SBS. The above is used to manufacture an asphalt mixture and evaluate its performance. They used 6 wt% of RA and 1 wt% of NS. They added, to the asphalt modified with RA and NS, 1, 2, and 3 wt% of SBS to improve the cracking resistance. Conventional characterization tests were performed on the modified asphalts. The manufacturing process of the asphalt mixture is not clear. A HWTD test (60 °C), low-temperature beam bending test ($-10\ °C$), moisture susceptibility test, FPB fatigue test, and aging resistance test were performed on the asphalt mixture. The modified asphalts obtained higher stiffness values. The results on the modified asphalt mixtures are consistent with the above, since rutting resistance increases. The additives improved the fatigue performance, water resistance, and aging resistance of the asphalt mixture. The SBS helped to improve the resistance to cracking at low temperatures. An economic analysis indicated that the modified asphalt mixture had higher cost-effectiveness.

Li et al. [62] modified an AC 73 pen asphalt binder with 10, 20, 30, 40, and 50 wt% BRA by a wet process in an on-site device in an asphalt plant. They performed conventional characterization tests on the modified asphalts. On the modified asphalt mixtures, they performed Marshall tests to measure dynamic and residual stability, freeze–thaw splitting tests to measure TSR, and low-temperature bending failure. The modified asphalt mixture significantly improved high-temperature resistance and aging resistance. In addition, the modified asphalt mixture was used in a maintenance project in the Lian City of Anhui Province. They used 30 wt% BRA to produce this mixture, as this was the optimum content obtained from the strength tests. They did not use a higher percentage because they observed segregation problems in the mixture with higher BRA contents. According to the researchers, the modified asphalt mixture had better application properties and workability and is suitable for large-scale production. As a limitation of the study, they report that the evaluation time of the test section was short.

Lv et al. [133] modified an AC 75.2 pen with BRA. Then, two other additives (SBR, nano-$CaCO_3$) were added to this modified asphalt, which was chosen to improve the low-temperature performance of the BRA-modified asphalt. Based on Marshall tests, they determined the optimum asphalt content of the control mixture (without BRA) and the BRA-modified mixture (4.2 and 4.4 wt% of the mixture, respectively). The BRA content was 3 wt% in the BRA-modified asphalt mixture. The ratio of BRA to base asphalt was 0.83:1. The SBR contents were 3 and 5 wt%, and the nano-$CaCO_3$ contents were 5 and 10 wt%. Initially, the AC was heated to 140 °C. When the base asphalt was melted, BRA was added, mixing at 140 °C for 10 min at 4000 rpm. Then, SBR and nano-$CaCO_3$ were added, respectively, to the asphalt, mixing for 25 to 30 min. They performed rheology tests with DSR and BBR. They also performed conventional characterization tests. The anti-rutting performance of the asphalt modified with 10% BRA/nano-$CaCO_3$ compound was the best. The proper addition of nano-$CaCO_3$ and SBR can improve thermal stability, with no negative effects on the aging resistance of BRA-modified asphalt, to a certain extent. The thermal cracking performance of 5% BRA/SBR-modified was effective, while that of nano-$CaCO_3$ is not obvious.

Huang et al. [134] modified an AC 71 pen with an RA and diatomite. The additions were 18 wt% RA, 13 wt% RA + 7 wt% diatomite, and 16 wt% RA + 9 wt% diatomite. Initially, the AC was preheated to 150 °C, and the RA and diatomite were added at room temperature. The mixture was heated and maintained at a temperature of 175 °C for 1 h and then mixed for 30 min at 3000 rpm. On the modified asphalts, they performed

rheological characterization using DSR and measured the viscosity. They also performed SEM visualizations on the RA and diatomite. The addition of RA and diatomite significantly increased the viscosity and stiffness of the base asphalt, improving rutting resistance. However, a decrease in fatigue performance was reported.

Fan et al. [135] modified an AC 68.2 pen with BRA (19, 39, 58, 77, 97 wt%) and SBR (2, 4, 5, 6, 8 wt%). The AC was initially mixed with the BRA at 165 °C for 30 min at 3000 rpm. They were then modified with SBS at 165 °C for 30 min at 3000 rpm. They performed conventional characterization tests, penetration at 15 °C, rheological characterization (DSR, BBR), and force ductility test. The BRA increased the stiffness of the base asphalt. BRA decreased the low-temperature performance of the binder (mainly attributed to the ash content); however, the addition of SBR can improve it.

Li et al. [136] modified two binders (AC 60 pen and AC 70 pen) with RA (2, 4, 6, 8 wt%). Initially, the ACs were heated to 140 °C. Then, the RA was added, mixing at 175–180 °C for 20 min. Finally, the mixture was stopped for 10 min and stirred again for 20 min. They performed conventional and rheological characterization (BBR) on the modified asphalts. They manufactured asphalt mixes with five gradations, controlling the temperature at 170–175 °C. They performed a mixed design by the Marshall method. The optimum RA content was 8 wt%. They performed uniaxial creep tests (15, 20, 35, 40, 60 °C), ITF tests (15 °C), low-temperature anti-cracking performance (−10 °C), and freeze–thaw split tests. RA can improve the rutting and aging resistance of the asphalt binder. However, it decreases the low-temperature performance. For the optimum content of 8 wt%, the modified mixes undergo good water and fatigue resistance (for areas with a minimum temperature of not less than −21.5 °C).

Qu et al. [137] modified an AC 68.2 pen with BRA (15 wt%). Two types of SBR (powder and latex) were then added to this modified asphalt. The BRA was crushed into fine particles that passed through the 0.15 mm sieve. The SBR contents were 2, 4, 6, and 8 wt%. SBR was mixed with AC and BRA at 165 °C and 3000 rpm for 30 min. They measured viscosity and performed rheological characterization (DSR and BBR), FTIR, and SEM observations. SBR improved the low-temperature performance of BRA-modified asphalt. Compared to latex SBR, powdered SBR significantly improved its high-temperature performance. They recommended the use of powdered SBR.

Lv et al. [138] modified a bio-asphalt (AC 60–80 pen with 5% contents of bio-oil residue distilled from vegetable oil) with BRA (5, 10, 15, 15, 20 wt%) to evaluate its aging resistance. Initially, AC and bio-oil were mixed at 150–160 °C for 30 min at 5000 rpm. The BRA was quickly added to the bio-asphalt and mixed at 150–160 °C for 60 min at 5000 rpm. Finally, this mixture was placed in an oven at 150 °C for 20 min. They simulated aging at TFOT and PAV. They performed rheological characterization (DSR and BBR), a temperature sweep test, a MSCR test at 58 °C, and SEM observations. BRA improved the high-temperature performance of bio-asphalt, and it is higher with increasing BRA. BRA-modified bio-asphalt is generally more sensitive to aging compared to base asphalt. The high-temperature properties of BRA-modified bio-asphalt with a BRA content of 20% are equivalent to neat asphalt. The bio-oil greatly improves the low-temperature rheological properties of BRA-modified asphalt. The BRA improves the sensitivity of bio-asphalt to aging under low-temperature conditions. The suggested optimum BRA content range is 15%–20%. BRA-modified bio-asphalt can meet the technical requirements of road engineering and has excellent aging resistance. The use of BRA and bio-oil can reduce the consumption of petroleum asphalt, drastically reduce the cost of materials, and produce ecological benefits.

Lv et al. [139] modified a bio-asphalt (AC 70.2 pen blended with bio-oil derived from the byproduct of trench oil in the process of rectifying biodiesel; 3–9%) with BRA (passing sieve #200; 5–20 wt%), optimizing its design through the Box-Behnken Design (BBD) method. Initially, the AC was placed in an oven at 135 °C for 2 h. Then, the bio-oil was slowly incorporated into the AC and stirred for 15 min, and subsequently mixed at 3000 rpm for 20 min. Afterward, BRA was incorporated into the bio-asphalt and stirred for

10 min, and subsequently mixed at 145 °C at 3000 rpm. As independent variables, bio-oil content, BRA content, and shear time were established. Conventional characterization, BBR, MSCR, and FTIR tests were performed on the modified bio-asphalt. Response surface method (RSM) and genetic algorithm optimization artificial neural network (GA-ANN) were used to analyze the behavior of the modified asphalt. The modified bio-asphalt performed better at low and high temperatures compared to neat asphalt. The optimum bio-oil content, BRA content, and shear time determined using the RSM and GA-ANN model were 6.3%, 11.2%, 52.8 min, 6.3%, 12.9%, and 76.6 min, respectively. Optimization with GA-ANN can further promote bio-oil and BRA recycling, save energy, and lead to more environmentally friendly construction material.

Li et al. [140] fabricated a FAM mixture with limestone as the aggregate, PG 64-22 as the asphalt binder, and RA from three different sources (Buton, QC, and UM). RA content was 8 wt%. Additionally, a 20% concentration of Buton RA was also prepared. Initially, the RA was mixed with the aggregate and asphalt binder and then placed in an oven at 135 °C for 2 h (STOA procedure). This mixture was compacted using the Superpave gyratory compactor (air void content varied from 1.57% to 2.52% by volume of the mixture). Then, the compacted samples were cut at both ends equally to obtain cylinders of 45 mm in height, extracting cylindrical specimens of 12.25 mm in diameter and 45 mm in height. The FAM mixtures were subjected to fracture tests (25 °C), fatigue damage (25 °C), and self-healing tests. RA significantly increased the fracture and fatigue resistance of the FAM mixture. UM and QC show better fatigue performance than Buton. RA QC exhibited the best self-healing properties followed by UM and Buton.

Li et al. [61] "activated" BRA samples (precipitated BRA after grinding—particles smaller than 3 mm—and heated at 150–180 °C for 9 min) and modified an AC 73 pen. The content of "activated" and non-"activated" BRA to modify the AC was 10, 20, 30, and 40 wt%. The modification process is not clear. They performed conventional characterization tests and AFM, and calculated the Derjaguin–Muller–Toporov (DMT) modulus. The modified asphalt with the "activation" process is stiffer than without the "activation" process. The DMT of the modified asphalt is approximately 2.5 times higher than that of the base asphalt.

Cheng et al. [141] modified an AC 87.65 pen (PG 64-22) with Xinjiang RA (XRA). The XRA contents were 8, 12, 16, and 20 wt%. The XRA particles exhibited sizes less than 1.18 mm. The modification process is not clear. They simulated aging at TFOT and PAV. They performed conventional characterization, FTIR, RV, dynamic shear oscillatory, frequency sweep, MSCR, and BBR tests. XRA helped to decrease the carbonyl functional group of the binders (increased aging resistance) and increased the PG at high service temperatures and the stiffness of the base asphalt (improved rutting resistance). However, high proportions of XRA could compromise the properties at low service temperatures. The optimum XRA dosage recommended by the authors was 12 wt%. At this percentage, the thermal stability, aging resistance, and low-temperature performance of the base asphalt are improved.

Yan et al. [142] modified a bean-based castor bio-asphalt with a European RA. The base asphalt used is an AC 62.8 pen. The content of castor beans is not clear. Initially, the base asphalt and bio-asphalt were preheated in a furnace at a constant temperature to melt both materials. RA was then added to the AC and mixed at 160 °C and 1500 rpm. Subsequently, the speed was increased to 3000 rpm and mixed for 60 min. Finally, the temperature was lowered to 140 °C to add the bio-asphalt and mixed at low speed for 30 min. To optimize the composition of the bio-asphalt and RA, they used a D-optimal mixture design (DMD). They performed conventional, rheological (DRS), and FTIR characterization. The addition of RA improved the performance of bio-asphalt at high temperatures. FTIR indicated that there was no chemical reaction between bio-asphalt, RA, and base asphalt. The effect of bio-asphalt and RA on the low-temperature performance of modified asphalt was not clear. Based on DMD, the following optimum compositions were obtained: 2.9% bio-asphalt, 16.7% RA asphalt, and 80.4% AC; 7.9% bio-asphalt, 6.3% RA and 85.8% AC.

Ren et al. [143] manufactured an asphalt mixture employing an AC 67 pen and BRA contents of 3, 3.5, and 4 wt%. The corresponding optimum asphalt contents of the mixture with BRA were 4.8, 4.8, and 4.9%, respectively. The manufacturing process of the asphalt mixtures is not clear. Only HWTD tests were performed. Five levels of soaking times (0, 10, 30, 330, 330, and 3600 min), four levels of temperature (40, 50, 60, and 70 °C) and three levels of BRA dosage (3%, 3.5%, and 4%) were selected. The rutting resistance of the RA mixture increases with increasing BRA dosages. However, with water intrusion, the rutting resistance weakens significantly.

Ren et al. [144] manufactured asphalt mixtures using an AC 67 pen, basalt aggregates, and BRA (1.5, 2.5, 3.5, 3.5, 4.5, and 4.5 wt% of the asphalt mixture). The mixing and compaction temperatures were 170 °C and 160 °C, respectively. However, the BRA addition process is not clear. They performed fatigue tests (strain-controlled) at 15, 40, and 60 °C and evaluated the combined effects of moisture (through water absorption ratio of 30, 50, 80, and 100%), high temperature, and stress level. The fatigue resistance of the BRA blend decreased with increasing moisture, temperature, and stress levels. However, the fatigue life of the asphalt mixture with BRA increased as the BRA content increased, but this growth trend is slower when the BRA content exceeds 3.5%. Based on economic criteria, they recommend 3.5% as the optimum BRA content.

Zhang et al. [145] fabricated four asphalt mastics with two asphalt binders (AC 65.3 and 68.9 pens) and two fillers: limestone and Iranian RA (10 wt% obtained based on previous studies). The above is to evaluate the influence of RA on the fatigue cracking resistance and self-healing characteristics of asphalt mastics. A weight ratio of filler to asphalt binder 1:1 was used. The manufacturing process of asphalt mastics is not clear. A frequency sweep test and fatigue-healing test were performed. RA could significantly increase the self-healing potential of asphalt mastic. Taking into account capillary diffusion theory, the addition of RA adversely affects the wetting mechanism of self-healing, but significantly improves the strength and efficiency of diffusive cohesive healing.

Yan et al. [146] modified an AC 69 pen with waste cooking oil—WCO (2, 4, 6 wt%) and European RA (6, 12, 18, 24 wt%). These modification percentages were chosen based on previous studies. Initially, the AC was heated to 160 °C to add the WCO by mixing at 1500 rpm for 10 min. Then, RA was added and the temperature was maintained between 150 and 160 °C for 30 min. Subsequently, the materials were mixed for 30 min at a speed of 3000 rpm at 160 °C. Finally, the mixture was mechanically stirred at 1000 rpm for 10 min. Conventional asphalt characterization, rheology (DSR, BBR), FTIR, and storage stability tests were performed. In addition, they performed economic and environmental analyses. WCO improves asphalt resistance to low-temperature cracking, while adverse effects on high-temperature performance can be compensated for by RA. The modified asphalt was less susceptible to temperature. Both RA and WCO are compatible with asphalt. They recommend 4% WCO and 18% RA to achieve better rutting, cracking, fatigue, and aging resistance performance. With these additive contents, the PG 64-22 base asphalt changed to PG 70-28. The addition of WCO and RA could reduce asphalt binder cost and contamination.

Wang and Xing [11] modified an AC 67.8 pen with a BRA (2, 3, and 4 wt%) containing 75% mineral powder and 25% pure natural asphalt (PNA). The percentages of BRA used generated modified asphalts with PNA contents of 10, 15, and 20%, respectively. Initially, the AC was heated to 150 °C, and then the BRA was added. The mixing of both materials was carried out at 165 °C for 30 min. Observations of the BRA were made by AFM and SEM. On the modified asphalt, they evaluated the viscosity and rheological properties (DSR, BBR). They manufactured a dense-graded AC-13 asphalt mixture with BRA-modified asphalt and a comparison mixture with SBS-modified asphalt. Shear strength, freeze–thaw splitting, and splitting fatigue (15 °C) tests were performed on the mixtures. The microstructural properties of PNA were more stable. The BRA minerals have a higher adsorption capacity than limestone mineral powder. The asphalt stiffened when using PNA, increasing the high-temperature PG from 64 to 70 °C when its content was increased from 10% to 20%

(increasing rutting resistance). However, PNA had a negative influence on low-temperature performance. The asphalt mixture with BRA showed better high-temperature performance, lower susceptibility to moisture, and higher resistance to fatigue cracking compared to the base asphalt mixture and the SBS-modified mixture.

Wen et al. [147] modified an AC 89 pen with RA, SBS, and CR. The modification contents studied were 2% SBS, 12% RA, 5% RA + 2% SBS, 10% RA + 14% CR, 10% RA + 12% CR, and 5% RA + 18 CR. AC and RA were mixed at 165 °C for 30 min at 2500 rpm. They simulated aging at TFOT and PAV. They performed rheology tests in DSR (temperature, linear amplitude, and time sweep test). The fatigue resistance of RA-modified asphalt is significantly affected by the type and amount of the modifier. Under controlled stress, the best fatigue response was obtained with 5% RA + 2% SBS-modified asphalt. Under controlled deformation, it was obtained with 5% RA + 18 CR. Reducing the RA content and increasing the CR content can significantly improve the fatigue resistance, extending its fatigue life by more than five times. Wen et al. [148] report a study similar to the one described above. They modified an AC 89 pen with AR contents of 5, 10, and 12 wt%. SBS contents of 1 and 2 wt% and CR contents of 14 and 18 wt% were added on top of the RA-modified asphalt. The AC and RA were mixed at 175 °C for 30 min at 2500 rpm. They performed conventional asphalt characterization. They simulated aging at TFOT and PAV to perform rheology tests at DSR (temperature and frequency sweep test, MSCR) and BBR. They also performed FTIR and FM tests. RA increased the stiffness and viscosity of AC, improving its performance at high temperatures. RA also shows good resistance to aging. SBS and CR help to improve the aging resistance and response of RA-modified bitumen at low temperatures. However, the performance grade of RA-modified asphalt was better when CR was used compared to SBS. The best performance is obtained when modified with 5% RA + 18% CR. At high service temperatures, the best performance is obtained when modified with 12% RA + 14% CR.

Liu et al. [149] modified an AC 74.5 pen with BRA (83 wt%) and SBR (3, 5, 7, 9 wt%). The three materials were mixed between 160 and 170 °C at 4500 rpm (the mixing time is not clear). They performed conventional asphalt characterization tests. They designed and manufactured an asphalt mixture by the Marshall method using asphalt modified with 5% SBR and 83% BRA. They also manufactured a mix with SBS-modified asphalt for comparison. High-temperature performance (60 °C), low-temperature performance (−10 °C), moisture stability, and FPB fatigue (15 °C) tests were performed on the mixtures. BRA + SBR improves the high-temperature performance of the blend and had a positive effect on moisture stability. BRA increases the fatigue resistance of the blend and SBR reduces the flexural tensile strength. SBR, however, improves the low-temperature anti-cracking resistance of BRA-modified asphalt. Asphalts modified with BRA + SBR have the same performance as asphalt modified with SBS.

Ma et al. [150] modified an AC 68 pen with RA (2, 4, 6, 8 wt%) from Sichuan, China. Initially, the AC was fluidized and mixed with the RA for 10 min. Then, the mixture was placed in an oven at 150 °C for 1 h. Finally, they were mixed at 150 °C for 30 min at 4000 rpm. They simulated aging at TFOT and PAV. They performed conventional characterization tests and rheology with DSR and FTIR. RA increased the viscosity and stiffness of the base asphalt. Aging resistance was better when the RA content was 6%. When RA was added, the molecular polarity was increased, improving the molecular association capacity and thermal stability of the asphalt system.

Yan et al. [151] modified an asphalt mixture with four combinations of asphalt modified with European RA and waste cooking oil (WCO) (18% RA + 0% WCO, 2% WCO + 18% RA, 4% WCO + 18% RA, and 4% WCO + 12% RA, by weight). An AC 69 pen was used as the base asphalt. Initially, the AC was preheated to 160 °C; later, the WCO was added and mixed at 1500 rpm for 10 min. At the same time, the RA was added. Then, the mixture was kept for 30 min at 150–160 °C. Subsequently, it was sheared at 3000 rpm for 30 min at 160 °C. Finally, air bubbles were removed by mechanical agitation at 1000 rpm for 10 min. Conventional characterization tests were performed on the modified asphalts.

Marshall tests (simulating LTOA aging), rutting test (60 °C), low-temperature indirect tensile testing (−10 °C), and moisture susceptibility tests were performed on the mixtures. The modified asphalts improved the low and high-temperature stability, water stability, and aging resistance of the asphalt mixture. The best performance of the mixtures was obtained when 4%WCO + 18%RA was used. RA has a significant effect on the stiffness increase of the base asphalt and asphalt mixture, while WCO can significantly improve its anti-cracking performance at low temperatures.

Su et al. [152] modified an AC 71.2 pen with BRA (10, 20, 30, 40 wt%) and investigated the influence of particle size and BRA content on the physical properties and stability of the modified asphalt. Initially, the BRA was kept in a box at 80 °C for 12 h to remove its moisture. Then, the BRA was ground into five different particle sizes (on average between 6.26 and 106.22 μm). The AC was heated to 155 °C and mixed with half of the BRA at 2500 rpm for 20 min. Then, the remaining BRA was added to the AC and mixing was continued at 2500 rpm for another 20 min. They performed conventional characterization tests and storage stability tests. They also evaluated viscosity–temperature susceptibility (VTS). The segregation of BRA in the modified asphalt during transportation and static storage processes was simulated and tested. The increase in stiffness and viscosity of the modified asphalt is directly proportional to the BRA content and inversely proportional to the particle size. Particle size and BRA content, as well as temperature and storage time, are positively correlated with the segregation of the modified bitumen mixture.

A summary of the review described in Section 4.3 is shown in Table 4.

**Table 4.** RA summary.

| Ref. | Asphalt Binder | RA Dosage (wt%) | Tests Carried out on Modified: | | Rutting | Fatigue | Resistance to: | | Moisture | Aging |
|---|---|---|---|---|---|---|---|---|---|---|
| | | | Asphalt Binder | Asphalt Mix | | | Temperature Cracking | | | |
| | | | | | | | Intermediate | Low | | |
| [11] | AC 67.8 pen | 2, 3, 4 | X | X | I | I | - | D | I | - |
| [59] | PG 64-22 asphalt | 8 to 20 | X | X | I | - | I | D | - | I |
| [60] | AC 87 pen | 5, 10, 15, 20 | - | X | I | I | - | D | I | - |
| [61] | AC 73 pen | 10, 20, 30, 40 | X | X | I | - | - | - | - | - |
| [62] | AC 73 pen | 10, 20, 30, 40, 50 | X | X | I | - | - | - | - | - |
| [65] | AC 70 pen | 5, 7.5, 10, 12.5 | X | X | I | - | - | - | I | - |
| [123] | AC 60 pen | 2, 4, 6, 8 | X | - | I | - | - | D | - | - |
| [124] | AC (170 Pa.s at 60 °C) | 20 | - | X | - | I | - | - | - | - |
| [125] | AC 60–70 pen | 9.5 wt% of the aggregate (SMA) | - | X | I | S | - | D | - | - |
| [126] | AC 70 pen | 20 | - | X | I | - | - | I | I | I |
| [127] | AC 70 pen | 20 | X | X | I | - | - | D | I | - |
| [129] | AC 67, AC 87 pen | 5, 10, 15, 20 | X | X | I | I | - | D | I | - |
| [130] | AC 70 pen | 2 | - | X | I | - | - | D | I | - |
| [131] | AC 60–70 pen | 2, 4, 6 | X | - | I | - | - | D | - | - |
| [132] | AC 60–70 pen | 6 | X | X | I | I | - | - | I | I |
| [134] | AC 71 pen | 13 to 18 | X | - | I | D | - | - | - | - |
| [135] | AC 68.2 pen | 19, 39, 58, 77, 97 | X | - | I | - | - | D | - | - |
| [136] | AC 60, AC 70 pen | 2, 4, 6, 8 | X | - | I | I | - | D | I | - |
| [137] | AC 68.2 pen | 15 | X | - | I | - | - | D | - | - |
| [138] | AC 60–80 pen + 5% bio-oil | 5, 10, 15, 15, 20 | X | - | I | - | - | - | - | I |
| [139] | AC 70.2 pen + 3–9% bio-oil | 5 to 20 | X | - | I | - | - | - | - | - |
| [140] | PG 64-22 | 8, 20 | - | X | - | I | - | - | - | - |
| [141] | AC 87.65 pen (PG 64-22) | 8, 12, 16, 20 | X | - | I | - | - | D | - | I |
| [143] | AC 67 pen | 3, 3.5, 4 | - | X | I | - | - | - | - | - |
| [144] | AC 67 pen | 1.5, 2.5, 3.5, 3.5, 4.5, 4.5 | - | X | - | I | - | - | - | - |
| [146] | AC 69 pen + 2, 4, 6%WCO | 6, 12, 18, 24 | X | - | I | I | - | - | - | I |
| [147] | AC 89 pen | 5 to 12 | X | - | - | I | - | - | - | I |
| [148] | AC 89 pen | 5 to 12 | X | - | - | I | - | - | - | I |
| [149] | AC 74.5 pen | 83 | - | X | I | I | - | D | I | - |
| [150] | AC 68 pen | 2, 4, 6, 8 | X | - | I | - | - | - | - | I |
| [152] | AC 71.2 pen | 10, 20, 30, 40 | X | - | I | - | - | - | - | - |

I: Increase; D: decrease; S: similar.

### 4.4. Oil Sands and Others

Anochie-Boateng and Tutumluer [79] studied the use of natural deposits of oil sands (with bitumen contents of 8.5, 13.3, and 14.5 wt%) as subgrade materials for temporary and permanent roads in oil sand fields. They used repeated load triaxial tests to characterize the oil sands. The oil sand samples were prepared using a gyratory compactor. They tested the samples under nine stress states (different confining pressures and cyclic deviatoric stress) applied at two temperatures (20 and 30 °C). The duration of the loading pulses was 0.1 and 0.5 s with rest periods of 0.9 and 0.5 s, respectively. They reported similar conclusions to those obtained in studies on conventional asphalt mixtures: (i) resilient modulus (RM) were generally higher at 20 °C than at 30 °C, and (ii) there was little or no statistically significant difference between RM values at 0.1 and 0.5 s loading pulse durations. They modified empirical regression models found in the literature for the case of unbound granular materials (K-theta, Witczak-Uzan, and the mechanistic–empirical pavement design guide—MEPDG models).

Kök et al. [153] used a NA composed of 17 wt% asphalt fraction and 83 wt% mineral fraction to modify an asphalt mixture. They used an AC 190 pen as an asphalt binder. They included 8.33% NA by the weight of the mixture. The process of NA inclusion is not clear. They performed ITSM (10, 20, and 30 °C), ITS, and dynamic creep (DC; 40 °C) tests. They reported an increase in stiffness under cyclic loading, resistance to moisture damage under freeze–thaw cycles, and resistance to permanent deformation. In addition, the NA-modified mixes decreased the optimum asphalt content in the mixture design by 1% (decreasing the production cost).

Olabemiwo et al. [154] modified a Nigerian NA (Agbabu, Ondo State) with PPA. According to the authors, Nigeria has a proven reserve of about 42.47 billion tons of bitumen (estimated to be the second largest in the world, but not yet explored for economic purposes). The proportion by weight of PPA added to the NA was 2, 4, and 6 wt%. The mixing of both materials was carried out at temperatures between 150 and 155 °C, 1200 rpm for 1 h. Conventional characterization tests of asphalt binders and FTIR were performed. They did not perform rheological characterization. PPA has the potential to improve the service life of NA. However, NA required a relatively higher concentration of PPA compared to some previous studies on other asphalts.

Olabemiwo et al. [155] modified Agbabu natural bitumen (ANB) with three polymers (HDPE, Polyethylene-co-vinyl-acetate—PEVA, and Polystyrene-co-butadiene—PSCB) at 2, 4, and 6 wt%. Initially, ANB was subjected to a process of extraction of moisture and purification. This is because ANB has a water content of 11.06%. Then, ANB was mixed with each polymer at 190–195 °C for 90 min at 1200 rpm. They thermally simulated the long-term aging of the modified samples (subjected to an oven at 60 °C for 3 weeks). FTIR and Oscillating Disc Rheometer (ODR) tests were performed. All three modifiers were able to reduce the thermal aging rate of the base binder. Samples modified with 2 and 6% PEVA showed better resistance to fatigue cracking, rutting, and aging.

Olabemiwo et al. [156] modified ANB (82 pen) with silver nanoparticles (AgNP) as antioxidant material at ratios of 1.5, 3.0, and 4.5 wt%. The crude ANB was dehydrated and purified and then modified in a stainless-steel reactor with AgNP via melt blend technique at a temperature of 120 °C under stirring at 1200 rpm for 1 h. They simulated long-term aging by subjecting the modified ANB to 60 °C for 3 weeks. They performed FTIR tests, conventional asphalts characterization, and ODR. FTIR analysis showed that the incorporation of AgNP decreased the carbonyl index value due to its antioxidant potential. AgNP increased the ignition point, kinematic viscosity, and stiffness of the base asphalt while decreasing its specific gravity. The ODR test showed that the modified ANB is less prone to fatigue cracking and rutting.

Hu et al. [157] modified an AC 63.5 pen with Selenice natural bitumen—SNB (5, 10, 15, 20, 25 wt%). The particle size of the finest SNB was about 1 mm. The SNB moisture content was 2.46%. For that reason, initially, it was preheated to remove the moisture. Mixing of the AC and SNB was carried out for 15 min at a temperature of 165 °C. Then,

it was sheared at 10,000 rpm for 45 min. For comparison purposes, they modified the AC with SBS (4 wt% of the asphalt modified with SNB). On the modified asphalts they performed conventional characterization, rheological-frequency, and temperature sweep, MSCR, ZSV, LAS, BBR, FM, FTIR, and GPC tests. They also manufactured an SMA with modified asphalt. An asphalt–aggregate ratio of 6.5% was adopted for virgin, SBS, and SNB-modified mixtures. Marshall, ITS, and rutting tests (60 °C) were performed on the asphalt mixture. The addition of SNB improved the water damage resistance and performance of the asphalt at high temperatures but reduced it at low temperatures. It also reduced the storage stability. They recommend an optimum SNB content of 15% (range 10–20%). They consider SNB to be a promising low-priced natural modifier with excellent rutting resistance properties.

## 5. Conclusions and Recommendations

Based on the bibliographical review, the following is concluded:

The most significant finding on which the different studies agree is that the best performance shown by NAs as binder and asphalt mixture modifiers is in high-temperature climates. They mainly increase viscosity, stiffness, and resistance to permanent deformation (rutting). This is mainly due to the high content of asphaltenes in NAs.

Another significant finding is that at low temperatures, NAs do not show good performance (mainly crack resistance decreases). However, under small addition contents, performance at low temperatures may not be compromised. Therefore, NAs have been combined in multiple studies with other modifiers such as SBS and SBR, among others.

Few studies have evaluated the performance of NA-modified binders and asphalt mixtures at intermediate service temperatures.

In chronological order, the studies have tried to improve performance in low-temperature climates (using polymers, WMA-additives, and other materials as modifiers). Another aspect that has evolved chronologically is the combination of NA with alternative materials such as bio-asphalts and industrial and construction wastes. The above were used to develop more environmentally friendly materials.

NAs show high compatibility with conventional asphalts. By modifying the latter, they tend to increase their resistance to aging and water and show high storage stability.

Workability is negatively affected by the incorporation of NA. For this reason, some studies have combined NA with additives used in WMAs, bio-asphalts, and waste cooking oil, among others.

It is unclear whether the increase in binder viscosity and stiffness is the result of the NA or the modification process (e.g., high mixing temperatures with the binder—average of 170 ± 10 °C). Likewise, greater viscosity is equivalent to higher mixing temperatures, hindering the process of manufacturing and construction and negatively impacting the environment. This last aspect has not been analyzed.

Most of the studies used NA as a modifier of asphalt binders, followed by asphalt mixtures. Few studies have been conducted using them as modifiers of asphalt mastics and porous and recycled asphalt mixtures.

Most researchers agree that NAs offer economic and environmental advantages, however, few studies have evaluated the economic and environmental impacts of using NAs on pavements.

Very few studies have conducted tests using Full-Scale Accelerated Pavement Testing (FS/APT).

NAs are more commonly used in industrial activities compared to use in road projects. However, the number of published studies using them as binder and asphalt mixture modifiers has increased in recent years.

The ranges of gilsonite, TLA, and RA most used in studies to modify asphalt binders are 1 to 24 wt%, 5 to 50 wt%, and 2 to 40 wt%, respectively.

In the future, the following are recommended: (i) conduct long-term studies (durability) and larger full-scale tests; (ii) further studies evaluating the chemical interaction

between the NA–asphalt binder-aggregate (physical-chemical and micro-structural performance); (iii) further studies evaluating the mastic performance; (iv) evaluate the use of NA in porous and recycled asphalt mixtures; (v) the influence of the binder–NA–aggregate interaction and asphalt mixture gradation; (vi) performance grade evaluation at intermediate temperatures of service; and (vii) environmental and socio-economic impacts evaluation.

**Author Contributions:** Conceptualization, H.A.R.-Q., C.A.Z.-M. and J.C.R.-C. methodology, H.A.R.-Q. and C.A.Z.-M.; validation and formal analysis, J.C.R.-C., H.A.R.-Q. and C.A.Z.-M.; resources, J.C.R.-C., H.A.R.-Q. and C.A.Z.-M.; writing—original draft preparation, H.A.R.-Q.; writing—review and editing, C.A.Z.-M., J.C.R.-C. and H.A.R.-Q.; funding acquisition, J.C.R.-C. All authors have read and agreed to the published version of the manuscript.

**Funding:** This research received no external funding.

**Institutional Review Board Statement:** Not applicable.

**Informed Consent Statement:** Not applicable.

**Data Availability Statement:** The datasets analyzed during the current study are available from the corresponding author upon reasonable request.

**Conflicts of Interest:** The authors declare no conflict of interest.

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
