# Peer review of "Natural Asphalts in Pavements: Review"

_sustainability, doi:10.3390/su15032098_

Round 1

Reviewer 1 Report

Title: Natural asphalts in pavements: review

Comments:

This paper did a comprehensive review work on the application cases of natural asphalts in pavement materials. Overall, the review paper shows a significant value by summarizing the interesting findings in previous studies. Here are some comments for further improve the paper:

* Most of the findings in previous studies are presented with text in this paper, which is not kind to reviewers and readers. Please display some representative results through citing the tables or figures.

* The English editing should be further improved throughout the whole manuscript.

* Both abstract and conclusion sections should be reorganized including the key points in this review paper.

* The structure (especially for the 3. Review part) should be reorganized through clear classifications with key points.

* In addition, some same reports from different previous papers should be merged and summarized rather than only listing the results.

Author Response

25-november, 2022

Dear Editor and Reviewers,

We would like to thank you for your valuable and insightful comments that have helped us to improve our manuscript. Please find also in the following paragraphs our answers to the comments. We have tried our best to clarify all the points raised. We hope that this new version of the manuscript is satisfactory for publication.

Reviewer 1

Comments:

This paper did a comprehensive review work on the application cases of natural asphalts in pavement materials. Overall, the review paper shows a significant value by summarizing the interesting findings in previous studies. Here are some comments for further improve the paper:

* Most of the findings in previous studies are presented with text in this paper, which is not kind to reviewers and readers. Please display some representative results through citing the tables or figures.

Answer: Thank you very much for your observation. You are right. For this reason, we made changes to the manuscript. The changes were as follows: i) the introduction is more concise and the old subchapter 1.2 (Natural Asphalts (NA)) was separated and placed as chapter 2; ii) chapter 3, now chapter 4, was divided into four subchapters (4.1 Gilsonite and asphaltites; 4.2 Trinidad Lake Asphalt (TLA); 4.3 Rock asphalt (RA); 4.4 Oil sands and others); iii) three tables were elaborated and introduced into the text (Tables 2, 3 and 4) to present a more reader-friendly summary.

* The English editing should be further improved throughout the whole manuscript.

Answer: Thanks for your comment. We had the document reviewed by an English expert.

* Both abstract and conclusion sections should be reorganized including the key points in this review paper.

Answer: Thank you very much for your observation. The conclusions were organized in order of importance to better present the most significant findings. We consider that the abstract is well described, for this reason we did not make changes. We hope your understanding about it.

* The structure (especially for the 3. Review part) should be reorganized through clear classifications with key points.

Answer: You are right. For this reason, we divided chapter 3 (now chapter 4) into four subchapters (4.1 Gilsonite and asphaltites; 4.2 Trinidad Lake Asphalt (TLA); 4.3 Rock asphalt (RA); 4.4 Oil sands and others).

* In addition, some same reports from different previous papers should be merged and summarized rather than only listing the results.

Answer: In order to take into account your observation, we have prepared a summary of the review described in Tables 2, 3 and 4.

Reviewer 2 Report

General Comments

The writing style and format (particularly section 1) are thesis style rather than a journal article. It should explain the background of a review paper on the topic, a summary of the literature, and the aim of this study without any subsection. The current form is missing the background, and the objective is unclear. Revise section 1.

Method

The method should explain the review, analysis, and classification process. Unfortunately, this form is quite unclear.

Review

·         This section is like each paragraph summarizing each paper, which is not the review. Instead, the review should be topic-based, not author based, that should compare and contrast the findings on that topic, called the critical review and analysis.

·         For example, the topic is the synthesis or physical properties, so the critical review focus on that. It should be supported with figures and tables to compare the findings or trends of others.

·         In the present form, it is not acceptable for a review paper.

Conclusion

After revising the structure of section 3 and performing the critical review, then the conclusion should include the significant finding of your critical analysis. For example, it could be the critical issues or significant benefits obtained through critical analysis.

Author Response

25-november, 2022

Dear Editor and Reviewers,

We would like to thank you for your valuable and insightful comments that have helped us to improve our manuscript. Please find also in the following paragraphs our answers to the comments. We have tried our best to clarify all the points raised. We hope that this new version of the manuscript is satisfactory for publication.

Reviewer 2

General Comments

The writing style and format (particularly section 1) are thesis style rather than a journal article. It should explain the background of a review paper on the topic, a summary of the literature, and the aim of this study without any subsection. The current form is missing the background, and the objective is unclear. Revise section 1.

Answer: Thank you very much for your comment. Based on your comment, the Introduction chapter was reorganized. The subchapters titles have been removed and the old subchapter 1.2 has been separated from the introduction and placed as chapter 2. Additionally, the objective was modified (see lines 51 to 55).

Method

The method should explain the review, analysis, and classification process. Unfortunately, this form is quite unclear.

Answer: Thank you very much for your comment. Based on your comment, the wording of the paragraph found in lines 175 to 182 was improved.

Review

This section is like each paragraph summarizing each paper, which is not the review. Instead, the review should be topic-based, not author based, that should compare and contrast the findings on that topic, called the critical review and analysis.

  • For example, the topic is the synthesis or physical properties, so the critical review focus on that. It should be supported with figures and tables to compare the findings or trends of others.
  • In the present form, it is not acceptable for a review paper.

Answer: Thank you very much for your comment. Based on your comment, the format of the review was changed. Chapter 3 (now chapter 4) was divided into four subchapters to improve the presentation ((4.1 Gilsonite and asphaltites; 4.2 Trinidad Lake Asphalt (TLA); 4.3 Rock asphalt (RA); 4.4 Oil sands and others). Additionally, three Summary Tables (Tables 2, 3 and 4) were prepared to make the review easier to understand.

Conclusion

After revising the structure of section 3 and performing the critical review, then the conclusion should include the significant finding of your critical analysis. For example, it could be the critical issues or significant benefits obtained through critical analysis.

Answer: Thank you for your comment. For us, the most significant finding found in the bibliographical review is that natural asphalts are recommended as modifiers for asphalt binders and asphalt mixes in high-temperature climates. Additionally, they are not recommended for low temperature climates. You can see these conclusions in lines 1341 to 1348.

Reviewer 3 Report

This manuscript provides a review of the sources of natural asphalts and their application in pavement materials (asphalt binders and mixtures). The effects of natural asphalts (e.g., the gilsonite, Trinidad Lake asphalts and rock asphalts) on the physical, chemical and rheological properties of the conventional asphalt binders or mixtures are focused on. The authors present detailed findings of the relevant literature in a chronological order. It is hoped that the following suggestions will be helpful in improving the manuscript.

(1) It is suggested that the Keywords be integrated, e.g., Trinidad Lake Asphalt and TLA essentially refer to the same natural asphalt.

(2) The significance and value of this work should be highlighted in an appropriate position and the Introduction is suggested to be presented in a more reasonable structure.

(3) The literature review was arranged in a chronological order. Could the authors clarify what regularity the relevant studies have shown over time?

(4) It is recommended that the authors distinguish between the utilization of natural asphalts in asphalt binders and mixtures in the manuscript.

(5) The authors provide a comprehensive and detailed summary of the available researches on natural asphalts, which is worthy of recognition. However, the literature review is arranged chronologically in an extremely long section and tends to present significant reading and comprehension barriers for the reader. In other words, the current format simply lists available research findings, making it difficult for the reader to understand the focus of the authors’ review. This section is proposed to be divided into several parts according to a specific logic.

(6) The significance of this literature review in guiding studies on the application of natural asphalts to pavements should be clarified. Also, further research recommendations were suggested to be distinguished from conclusions.

Author Response

25-november, 2022

Dear Editor and Reviewers,

We would like to thank you for your valuable and insightful comments that have helped us to improve our manuscript. Please find also in the following paragraphs our answers to the comments. We have tried our best to clarify all the points raised. We hope that this new version of the manuscript is satisfactory for publication.

Reviewer 3

This manuscript provides a review of the sources of natural asphalts and their application in pavement materials (asphalt binders and mixtures). The effects of natural asphalts (e.g., the gilsonite, Trinidad Lake asphalts and rock asphalts) on the physical, chemical and rheological properties of the conventional asphalt binders or mixtures are focused on. The authors present detailed findings of the relevant literature in a chronological order. It is hoped that the following suggestions will be helpful in improving the manuscript.

  • It is suggested that the Keywords be integrated, e.g., Trinidad Lake Asphalt and TLA essentially refer to the same natural asphalt.

Answer: Thank you very much for your comment. Trinidad Lake Asphalt and TLA were integrated (see line 20).

  • The significance and value of this work should be highlighted in an appropriate position and the Introduction is suggested to be presented in a more reasonable structure.

Answer: Thank you very much for your comment. Considering his observation, the Introduction chapter was restructured. The subchapter tittles have been removed and the old subchapter 1.2 has been removed from the introduction and placed as chapter 2.

  • The literature review was arranged in a chronological order. Could the authors clarify what regularity the relevant studies have shown over time?

Answer: The clear conclusion of the review carried out is that NA are materials that improve the performance of asphalt binders and asphalt mixtures in high-temperature climates. However, at low temperatures this performance decreases. For this reason, in chronological order, the studies have evolved to try to improve performance in low-temperature climates (using polymers, WMA-additives and other materials as modifiers). Another aspect that has evolved chronologically is its use in asphalt mixes that use alternative materials such as bio-asphalts, and industrial and construction wastes. The above to develop more environmentally friendly materials. This conclusion was written in the conclusions (see lines 1351 to 1355).

  • It is recommended that the authors distinguish between the utilization of natural asphalts in asphalt binders and mixtures in the manuscript.

Answer: Thank you very much for your comment. You're right. For this reason, three tables were prepared (Tables 2, 3 and 4), and there it is mentioned in which cases natural asphalts were studied as modifiers of asphalt binders and/or asphalt mixtures.

  • The authors provide a comprehensive and detailed summary of the available researches on natural asphalts, which is worthy of recognition. However, the literature review is arranged chronologically in an extremely long section and tends to present significant reading and comprehension barriers for the reader. In other words, the current format simply lists available research findings, making it difficult for the reader to understand the focus of the authors’ review. This section is proposed to be divided into several parts according to a specific logic.

Answer: Thank you very much for your comment. You're right. For this reason, we divided chapter 3 (now chapter 4) into four subchapters (divided more logically by type of natural asphalt; 4.1 Gilsonite and asphaltites; 4.2 Trinidad Lake Asphalt (TLA); 4.3 Rock asphalt (RA); 4.4 Oil sands and others).

(6) The significance of this literature review in guiding studies on the application of natural asphalts to pavements should be clarified. Also, further research recommendations were suggested to be distinguished from conclusions.

Answer: Thanks for the observation. Based on your comment, we added the word "recommendations" to the title of chapter 5. These recommendations are presented in the last paragraph (lines 1379 to 1385).

Reviewer 4 Report

This study reviewed the use of natural asphalts in pavement applications. The literature review is comprehensive, and the authors’ efforts are greatly appreciated.  

The introduction, methods, and conclusions are well-written. My comment is mainly about the section 3 review. It is a bit difficult for me to follow and seems to need more coherent logic. The current version lists relevant studies paragraph by paragraph. I suggest the authors divide section 3 into several subsections by certain logic, for example, the different types of use of NA, or the effects on different pavement performances (high-temperature or low-temperature, etc.), or something else, to make it easier for readers to follow. Also, for literature review, it would be great if you could make connections and comparisons between different studies, instead of listing studies one by one.

Author Response

25-november, 2022

Dear Editor and Reviewers,

We would like to thank you for your valuable and insightful comments that have helped us to improve our manuscript. Please find also in the following paragraphs our answers to the comments. We have tried our best to clarify all the points raised. We hope that this new version of the manuscript is satisfactory for publication.

Reviewer 4

The introduction, methods, and conclusions are well-written. My comment is mainly about the section 3 review. It is a bit difficult for me to follow and seems to need more coherent logic. The current version lists relevant studies paragraph by paragraph. I suggest the authors divide section 3 into several subsections by certain logic, for example, the different types of use of NA, or the effects on different pavement performances (high-temperature or low-temperature, etc.), or something else, to make it easier for readers to follow. Also, for literature review, it would be great if you could make connections and comparisons between different studies, instead of listing studies one by one.

Answer: Thank you for your comment. You are right. Chapter 3 (now chapter 4) was modified and divided into four subchapters (4.1 Gilsonite and asphaltites; 4.2 Trinidad Lake Asphalt (TLA); 4.3 Rock asphalt (RA); 4.4 Oil sands and others). Additionally, summaries of each subchapter are presented in three tables (Tables 2, 3 and 4) which were prepared for this purpose.

Round 2

Reviewer 1 Report

Most of my comments have been addressed well.

Author Response

12-december, 2022

Dear Editor and Reviewers,

We would like to thank you for your valuable and insightful comments that have helped us to improve our manuscript. Please find also in the following paragraphs our answers to the comments. We have tried our best to clarify all the points raised. We hope that this new version of the manuscript is satisfactory for publication.

Reviewer 1

Most of my comments have been addressed well.

Answer: Thank you very much for the review. Now our manuscript is much better thanks to your comments and observations.

Reviewer 2

Much improved than the earlier version no further comments.

Answer: Thank you very much for the review. Now our manuscript is much better thanks to your comments and observations.

Reviewer 3

The authors have revised the manuscript accordingly. The revised draft has responded to the questions and suggestions raised. The full text is more precise than the original. So the manuscript is recommended to be accepted.

Answer: Thank you very much for the review. Now our manuscript is much better thanks to your comments and observations.

Reviewer 4

The added tables are nice. Based on that you can do more analyses, for example, commenting on what the main problems are, what the consensus is, what is in debate, what problems need further study, etc. Make comparisons between different papers, not just list them one by one. This kind of analysis composes the merit of a literature review. To me, listing references paragraph by paragraph is not a good way to write a literature review.

Answer: Thank you very much for you comment. Based on the Review Report (Round 1), we divided the results chapter into four subchapters (4.1 Gilsonite and asphaltites; 4.2 Trinidad Lake Asphalt (TLA); 4.3 Rock asphalt (RA); 4.4 Oil sands and others). Additionally, three tables were elaborated and introduced into the text (Tables 2, 3 and 4) to present a more reader-friendly summary. Based on the above, the conclusions were organized in order of importance to better present the most significant findings. The clear conclusion and consensus of the review carried out is that NA are materials that improve the performance of asphalt binders and asphalt mixtures in high-temperature climates. However, at low temperatures this performance decreases. For this reason, in chronological order, the studies have evolved to try to improve performance in low-temperature climates (using polymers, WMA-additives and other materials as modifiers). Another aspect that has evolved chronologically is its use in asphalt mixes that use alternative materials such as bio-asphalts, and industrial and construction wastes. The above to develop more environmentally friendly materials. Also based on these tables and the main problems, we make recommendations for future work (lines 1379 to 1385). For us, listing the references chronologically and making a brief description of them was a didactic way of showing the review. Perhaps it is not the best, but we believe that showing each manuscript reviewed in summary was a good option. We hope for your understanding.

Reviewer 2 Report

Much improved than the earlier version no further comments

Author Response

(The authors gave the same response as above.)

Reviewer 3 Report

The authors have revised the manuscript accordingly. The revised draft has responded to the questions and suggestions raised. The full text is more precise than the original. So the manuscript is recommended to be accepted.

Author Response

(The authors gave the same response as above.)

Reviewer 4 Report

The added tables are nice. Based on that you can do more analyses, for example, commenting on what the main problems are, what the consensus is, what is in debate, what problems need further study, etc. Make comparisons between different papers, not just list them one by one.

This kind of analysis composes the merit of a literature review. To me, listing references paragraph by paragraph is not a good way to write a literature review.

Author Response

(The authors gave the same response as above.)

Round 3
